# Densely vascularized thick 3D tissue shows enhanced protein secretion constructed with intermittent positive pressure

Misako Katsuura [1,2], Jun Homma [1] ✉, Yuhei Higashi[3], Hidekazu Sekine [1] ✉ & Tatsuya Shimizu [1]

Constructing a dense vascular endothelial network within engineered tissue is crucial for successful engraftment. The present study investigated the effects of air-compressing intermittent positive pressure (IPP) on co-cultured mesenchymal stem cells and vascular endothelial cells and evaluated the potential of IPP-cultured cell sheets for transplantation therapy. The results demonstrated that the IPP (+) group exhibited a denser vascular endothelial network and significantly increased cell sheet thickness compared to the IPP (-) group. Furthermore, in vivo experiments showed that IPP-cultured cell sheets enhanced the secretion of Gaussian luciferase by genetically modified mesenchymal stem cells. These findings highlight the IPP method as a technique that simultaneously enables the thickening of planar tissues and the construction of vascular networks. This approach demonstrates promise for fabricating functional, transplantable, and thick tissues with dense vascularization and a high capacity for protein secretion, paving the way for novel applications in regenerative medicine.

Organ transplantation from a donor represents a radical treatment for a number of refractory diseases. However, organ transplantation therapy faces a challenge in becoming a more widely adopted therapy due to the organ donor shortage. Regenerative medicine using tissue engineering is anticipated to be a promising approach for treating refractory diseases. The primary objective of tissue engineering is to construct various tissue types from cells and materials, with the aim of replacing or enhancing the functionality of a given biological organism[1]. A cell sheet is a tissue that maintains intercellular connections and adhesion factors, making it a highly effective methodology for constructing complex three-dimensional tissues[2]. However, the construction of thick tissues cannot be achieved by simply stacking multiple layers of cell sheets in rapid succession. The difficulty arises because the central part of the tissue tends to suffer from hypoxia and malnutrition. Previously, we reported on the successful construction of thick tissue using a multi-step method, in which cell sheets were layered and transplanted every few days. This process involved the introduction of a vascular network into the cell sheets prior to their transplantation[3]. Thus, the provision of a dense vascular endothelial network produced in advance was proven to be an effective method for enhancing the efficacy of cell sheet-based tissue construction[3,4].

The challenge in the construction of a dense vascular endothelial network in vitro is the lack of chemical and mechanical stresses naturally present in the body's internal environment. Several methods have been reported to promote vascular endothelial networks in vitro. Currently, one of the most effective methods to promote vascular endothelial network formation in culture is the addition of pro-angiogenic factors such as vascular endothelial growth factor (VEGF)[5] and fibroblast growth factors (FGFs)[6] are added in the culture medium. Mesenchymal stem cells (MSCs) are also known to secrete various pro-angiogenic factors, which may lead to paracrine effects[7]. Moreover, it has been reported that the co-culture of MSCs and vascular endothelial cells (vECs) can form a dense vascular endothelial network[8]. Furthermore, the extracellular matrix (ECM), including collagen, plays a crucial role in supporting cell adhesion, survival, and migration, thereby providing an optimal environment for the construction of vascular endothelial networks[9]. Several studies have demonstrated that MSCs secrete various ECMs, which have been linked to tissue regeneration, self-renewal, and stem cell differentiation, making them useful for tissue engineering[10]. Among MSCs, adipose tissue-derived MSCs (ASCs) have been reported to exhibit superior angiogenic effects[11]. Therefore, with the goal of promoting a vascular endothelial network in vitro, co-culturing MSCs, particularly ASCs, with vECs is presumed to be an effective approach to promoting pro-angiogenic factors and ECMs.

[1]Institute of Advanced Biomedical Engineering and Science, TWIns, Tokyo Women's Medical University, Tokyo, Japan. [2]Department of Pediatrics, Tokyo Women's Medical University School of Medicine, Tokyo, Japan. [3]Tokaihit Co. Ltd, Shizuoka, Japan. ✉e-mail: homma.jun@twmu.ac.jp; sekine.hidekazu@twmu.ac.jp

Moreover, recent reports indicate that mechanical stimulation, such as shear stress, hydrostatic pressure, and stretch stress, on vECs in vitro changed the morphology and function of cells[12–14]. Shear stress is the most studied mechanical stimulation on vECs for promoting vascular endothelial network[15]. The application of mechanical stimulation in vitro typically necessitates the construction of a complex system, which involves the assembly of various device components and their adjustment according to the size of the cell culture dish. We have developed an in vitro pressurizing system that enables the easy application of external positive pressure. Some techniques have already been described for applying pressure to cells in vitro. These include the application of mechanical stimuli to cells by deforming the bottom of a dish made of soft materials, the alteration of the height of the culture medium in a culture dish, and the compression of the volume of the medium[13,16,17]. In contrast, our system employs the application of pressure to the cells through the movement of air in and out of the chamber[18]. The distinguishing feature of our system is that it is possible to pressurize the object inside the chamber. Furthermore, by changing the chamber size, it can be customized to a variety of forms, from an extracted organ to cells on a culture dish. In addition to its simplicity, it is also characterized by its ability to quickly switch pressure. The system is air-pressurized, and the chamber can be rapidly depressurized by switching from a closed to an open system. This enables the application of an intermittent positive pressure (IPP) to cells and organs. While the effects of IPP on organ perfusion have been previously reported[18], its effects on cells have not been thoroughly investigated.

The aim of the present study was to investigate the effects of IPP culture on co-cultured MSCs and vECs in two different environments: a planar environment and a cell sheet environment, and then to demonstrate the potential of IPP-cultured cell sheets for transplantation treatment. Demonstrating the effects of this pressure device on cells is a step towards better understanding the effects of mechanical stimulation on cells in order to help create functional tissues.

The present study demonstrated that IPP culture produced cell-dense and rich ECM tissues. Furthermore, IPP promoted the formation of vascular endothelial networks in vitro. Transplantation experiments suggest a potential improvement in protein secretion into the systemic circulation under IPP conditions. Finally, we found that IPP enhanced the insulin secretory function of pancreatic islet β cells in tri-cultured conditions with MSCs and vECs. These findings indicate that IPP is effective in improving cellular function and suggest its potential use in constructing a high-density vascular endothelial network in three-dimensional implantable tissues.

## Results

### Impact of intermittent positive pressure on endothelial cells forming vascular networks in planar co-culture conditions

An experiment was conducted to investigate the effect of IPP on co-cultured human adipose-derived stem cells (hASCs) and human umbilical vein endothelial cells expressing green fluorescent protein (GFP-HUVECs) in a planar environment (Fig. 1a). Over a period of three days, GFP-HUVECs infiltrated the intracellular spaces and began to connect with each other, forming a vascular network in both the IPP (−) and (+) groups (Fig. 1b, c). The number of network junctions did not differ significantly between the IPP (−) and IPP (+) groups [IPP (−) vs IPP (+) (count/mm²); 7.1 ± 4.1 vs 9.4 ± 2.7, $p = 0.10$, Cohen's $d = 0.66$] (Fig. 1d), and the total network length in the IPP (+) was also not significantly different [IPP (−) vs IPP (+) (mm/mm²); 3.5 ± 1.2 vs 4.2 ± 0.7, $p = 0.09$, Cohen's $d = 0.90$] (Fig. 1e).

### Applying intermittent positive pressure to the cells induced an aerobic condition and a proliferative state

The pressurization system employed a mechanism that forcefully introduced air into the chamber, effectively pressurizing it. Then, applying IPP to the culture medium would affect the level of dissolved oxygen, and then the medium conditions would change the cell metabolism. So, cellular metabolism was examined with a focus on lactate, a product of anaerobic metabolism. On day 3, we evaluated the lactate/glucose (L/G) ratio in the planar co-culture environment under IPP (−) and IPP (+) (Fig. 1a). The L/G ratio represents the ratio of lactate production to glucose consumption in the medium and is a measure of the aerobic status and anaerobic respiration[19]. The L/G ratio was lower in IPP (+) [IPP (−) vs IPP (+); 2.0 ± 0.2 vs 1.8 ± 0.2] than in IPP (−) (Fig. 1f). Then, the total cell number in the dish under IPP (+) was higher than IPP (−) [IPP (−) vs IPP (+) (cells/mm²); 1190.5 ± 311.7 vs 1388.3 ± 257.6] (Fig. 1g). The low L/G ratio, despite the high total cell number, indicates that applying IPP to the cells is associated with the development of aerobic conditions in the cells. The proportion of GFP-HUVECs among all cells showed no significant difference between the IPP (−) group (7.4 ± 3.4%) and the IPP (+) group (7.1 ± 3.7%) (Fig. 1h).

### A cell sheet environment established vascular endothelial networks more effectively than a planar culture environment

A cell sheet is a tissue with a higher density than a planar culture dish[20]. When cells are seeded on a temperature-responsive culture dish and harvested as a cell sheet, the cells that were adhered to the dish begin to shrink as they detach. Immediately after detachment (Supplementary Fig. 2a), the cell sheet exhibits an area that is 12 times smaller than in the planar culture state (Supplementary Fig. 2b). As a result, the cell sheet tissue becomes thicker and easier to handle as a planar cell tissue. Since we were using cell sheets for transplantation applications, the subsequent experiments were conducted using cell sheets. At first, we conducted a comparative analysis of the expression of angiogenic factors in hASC mono-culture conditions, in both a cell sheet environment and in a planar environment (Supplementary Fig. 1a). In the comparison of gene expression in hASC mono-culture, angiogenic factors such as *VEGFA*, angiogenin (*ANG*) and hypoxia-inducible factor 1 subunit alpha (*HIF1A*) were particularly elevated in the cell sheet environment (Supplementary Fig. 1b). A promotion of the vascular endothelial network of the co-cultured vECs was more pronounced in the cell sheet environment than in the planar environment (Supplementary Fig. 2c). On day 3, the vascular endothelial network was compared between planar cultures and cell sheets (Supplementary Fig. 2d, e). As co-cultured cell sheets shrink immediately after detachment from the temperature-responsive culture dish, the analysis was conducted using the unit area "per mm²" to account for differences in cell density between the two groups. Both the number of network junctions and the length of the entire network were significantly greater in the cell sheet environment than in the planar culture conditions [cell sheet vs planar culture: (count/mm²); 12.4 ± 3.2 vs 128.1 ± 29.3 (Supplementary Fig. 2d) (mm/mm²); 3.9 ± 0.5 vs 17.2 ± 1.8 (Supplementary Fig. 2e)]. Furthermore, by day 5, the vascular network in planar cultures had deteriorated, whereas the vascular network in the cell sheet environment was well-maintained (Supplementary Fig. 2c). These results demonstrated that the cell sheet environment facilitated the formation of a dense vascular endothelial network and contributed to its sustained maintenance over 5 days.

### Vascular endothelial cells in a co-cultured cell sheet environment under intermittent positive pressure formed developed vascular networks more than under non-pressurization

The aim of this experiment was to investigate the effect of IPP on the formation of a vascular endothelial network of co-cultured vECs in a cell sheet environment (Fig. 2a) because we have shown that the cell sheet is more likely to construct a dense vascular endothelial network. Furthermore, to observe the prolonged effects of IPP, we extended the evaluation period to 5 days, during which the vascular network can be maintained in the cell sheet environment. During the culture, vECs formed a network, and the cells of IPP (+) formed a denser vascular endothelial network (Fig. 2b, c). The number of network junctions and total network length was much greater in IPP (+) than IPP (−) [IPP (−) vs IPP (+): (count/mm²); 62.4 ± 24.7 vs 82.7 ± 19.8 (Fig. 2d) (mm/mm²); 11.2 ± 2.1 vs 13.7 ± 1.8] (Fig. 2e), which demonstrated that IPP was associated with the formation of a dense vascular endothelial network.

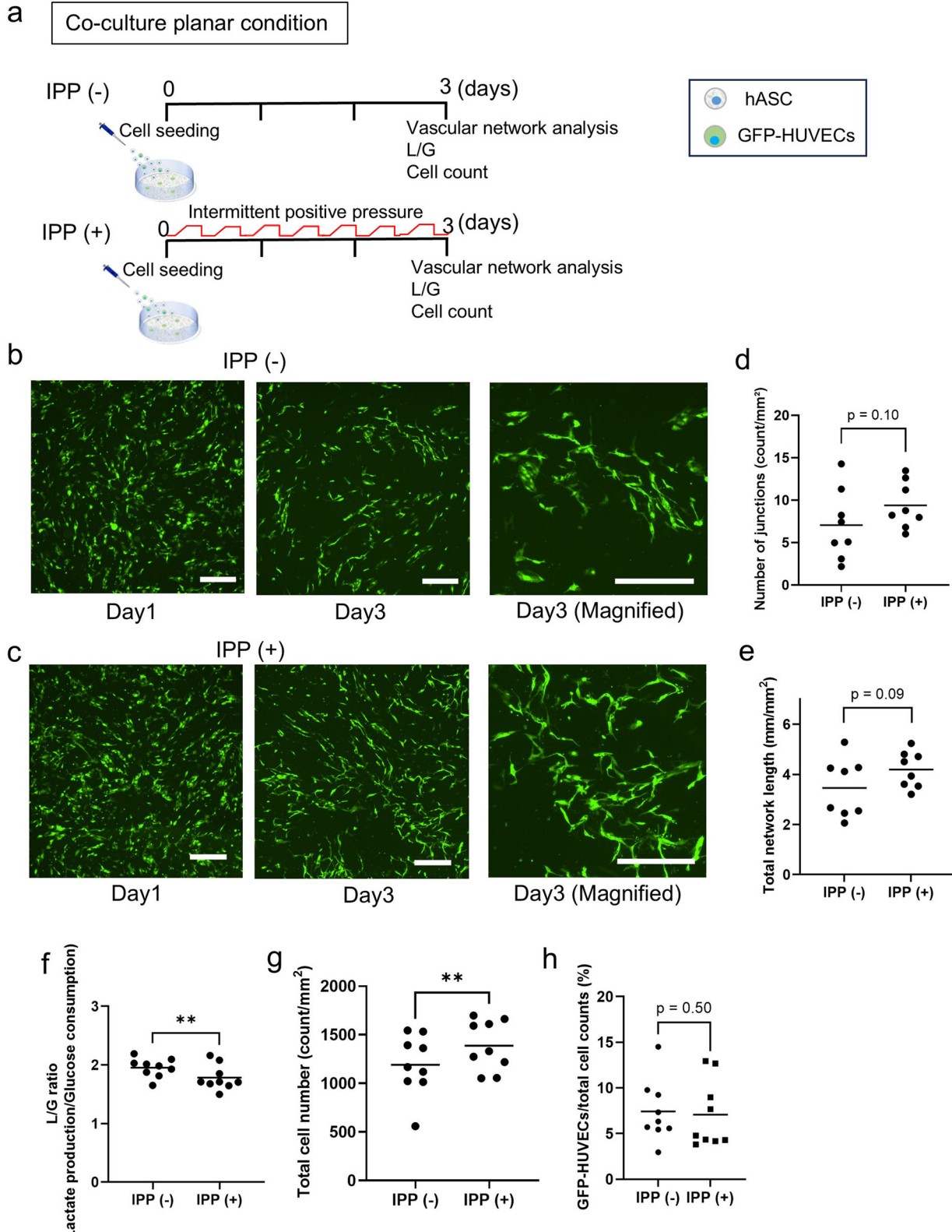

**Culturing the cell sheets under intermittent positive pressure resulted in morphological changes**

The next experiment was conducted to investigate any morphological changes in co-cultured cell sheets due to the effect of IPP (Fig. 2a). Upon scanning with the 3D-OCT system, the cell sheet under the IPP (+) showed more uniform thickening compared to the IPP (−) (Fig. 3a). Then, OCT images were analyzed and the thickness of the cell sheet attached to the dishes was significantly thicker in the IPP (+) group [IPP (−) vs IPP (+) (µm); 36.4 ± 4.0 vs 45.7 ± 8.3] (Fig. 3b). A comparative analysis of the thickness of the cell sheets in hematoxylin-eosin (HE) stained sections showed that the cell sheets in the IPP (+) group were thicker than in the IPP (−) group [IPP (−) vs IPP (+) (µm); 20.0 ± 6.2 vs 23.3 ± 6.4] (Fig. 3c, e).

**Fig. 1 | Comparison in planar co-cultured condition between applied intermittent positive pressure and non-pressurization.** **a** Diagram of the co-cultured planar condition experiment. **b** Representative image of GFP-HUVECs (*green*) under intermittent positive pressure (IPP) (−) on day 1 (*left*) and day 3 (*middle, right*). Scale bar: 500 μm. **c** Representative image of GFP-HUVECs (*green*) under IPP (+) on day 1 (*left*) and day 3 (*middle, right*). Scale bar: 500 μm. **d** The total number of endothelial network junctions on day 3 (*n* = 8, measured from 8 independent culture dishes, with the mean of 3 randomly selected fields of view per dish, *p* = 0.10, Cohen's *d* = 0.66). **e** The length of the endothelial network on day 3 (*n* = 8, measured from 8 independent culture dishes, with the mean of 3 randomly selected fields of view per dish, *p* = 0.09, Cohen's *d* = 0.90). **f** The ratio of lactate production to glucose consumption (L/G ratio) on day 3 (*n* = 9, independent samples, \*\**p* < 0.01, Cohen's *d* = 0.90). **g** The total number of the cells on day 3 (*n* = 9, independent samples, \*\**p* < 0.01, Cohen's *d* = 0.69). **h** The percentage of GFP-HUVECs of the total cell number in the co-cultured cells in planar conditions (*n* = 9, independent samples, *p* = 0.50, Cohen's *d* = 0.10). **d–h** Plots in the graph show the value of each experiment and the bar shows the average. L/G: The ratio of lactate production to glucose consumption.

This thickness was less than that obtained in 3D-OCT with the dehydration treatment. These results suggest that IPP contributed to the formation of a thicker organization of co-cultured cell sheets.

### The cell sheet cultured under intermittent positive pressure consisted of a greater amount of collagen and number of cells

To elucidate the factors contributing to the observed thickening of the cell sheet in the IPP (+) group, experiments were carried out to investigate compositional changes (Fig. 2a). Observation of Sirius red-stained sections showed collagen fiber had filled in the cell sheets under both conditions (Fig. 3d). To quantify the amount of collagen, we sonicated the cell sheets and measured the amount of collagen on day 5 (Fig. 2a). The quantification of the amount of collagen revealed that the cell sheets in the IPP (+) group contained more collagen than those in the IPP (−) group [IPP (−) vs IPP (+) (μg/sheet); 9.1 ± 2.0 vs 11.0 ± 2.3] (Fig. 3f). The number of cells in the IPP (+) cell sheet group was greater than that in the IPP (−) group. The cell count on day 0 was $5.0 ± 1.1 × 10^5$/sheet (*n* = 5). On day 5, cell count was $3.6 ± 0.8 × 10^5$/sheet in the IPP (−) group and $4.3 ± 0.8 × 10^5$/sheet in the IPP (+) group (Fig. 3g). The results indicate that the decline of cell numbers was suppressed under the IPP (+) condition. The proportion of GFP-HUVECs in all cells was no significant differences between in the IPP (+) group (6.2 ± 1.0%) compared to the IPP (−) group (4.3 ± 1.5%) (Fig. 3h). The cell viability of the total cells on day 5 was significantly higher in the IPP (+) group (85.0 ± 7.1%) compared to the IPP (−) group (69.1 ± 7.3%) (Fig. 3i). On the other hand, the forward scatter (a measure of cell size in flow cytometry based on the light scattered in the forward direction as cells pass through a laser beam), did not change in the two conditions, suggesting that the size of the cells was not changed by IPP culture [hASCs: IPP (−) vs IPP (+); 293.0 ± 34.3 vs. 291.6 ± 18.4 (Fig. 3j; *left*), GFP-HUVECs; 231.8 ± 91.6 vs. 213.7 ± 123.4 (Fig. 3j; *right*)]. The morphological change observed was a thicker cell sheet in the IPP (+) group due to an increase in collagen and cell viability, rather than the size of each cell.

### Intermittent positive pressure increased the air saturation at the bottom of the cell sheet attached to the dish

As shown in Fig. 1f, g, this mechanism for applying pressure appears to increase the level of dissolved oxygen in the medium. So, to evaluate the dissolved oxygen at the bottom of the dish where it meets the cell sheet, the air saturation in the medium was measured (Fig. 4a). In both conditions, the relative ratio of air saturation gradually decreased over time. However, upon reaching equilibrium, the IPP (+) exhibited a consistently higher air saturation than the IPP (−) (Fig. 4b). The experiment was conducted three times, and the same phenomenon was observed in all measurements, confirming its reproducibility (Fig. 4c). The results shown in Fig. 1f, g and Fig. 4 provide support for the hypothesis that the IPP increases the dissolved oxygen in the medium, thereby enabling the cells to be cultured under aerobic respiration conditions.

### Comparison of RNA expression related to the promotion of vascular endothelial network formation in co-cultured cell sheets under intermittent positive pressure and non-pressurization

In order to gain insight into the differences in vascular endothelial network promotion between the IPP (−) and IPP (+) from the perspective of gene expression, total RNA sequencing of the co-cultured cell sheet was performed (Fig. 2a). In the present study, the results were confirmed in tests of mechanotransduction, cell cycle, ECM, Hypoxia-inducible factor (HIF) family and angiogenesis paracrine factors. Figure 5a shows the expression of certain RNA related to mechanotransduction[21,22]. The genes encoding Notch receptor 1 (*NOTCH1*), Notch receptor 3 (*NOTCH3*), and Notch receptor 4 (*NOTCH4*), which are expressed in vECs and associated with shear stress[23], exhibited increases of 0.6-, 0.5-, and 0.3-Log2 fold changes, respectively. Additionally, the P2X4 receptor (*P2RX4*), an ion channel that senses stimulation through mechanotransduction caused by shear stress[24], demonstrated an increase of 0.4-Log2 fold change in the IPP (+) group. Furthermore, genes related to downstream signaling of mechanotransduction under shear stress, including the VE-cadherin gene (*CDH5*), the VEGFR2 gene (*KDR*), and the eNOS gene (*NOS3*) showed increased expression in the IPP (+) group (0.9-, 0.6-, 1.0- and 1.0-Log2 fold changes, respectively) (Fig. 5a). These results suggest that IPP caused mechanotransduction stimuli to the cells. Figure 5b shows the expression of certain RNA associated with cell cycle regulation. The RNA expression of the MYC gene (*MYC*), cyclin D1 gene (*CCND1*), cyclin-independent kinase 6 gene (*CDK6*), and cyclin A1 gene (*CCNA1*), all of which are related to cell cycle progression and cell proliferation in mesenchymal stem cells[25], were increased in the IPP (+) group (0.5-, 0.3-, 0.3- and 1.4-Log2 fold changes, respectively). As these genes are downstream targets of Notch receptor signaling, the results suggest a potential link between IPP and enhanced cell proliferation. Figure 5c shows the expression of certain RNA related to ECMs. The RNA expression of genes related to collagen type 1 (*COL1A1*), the major type of collagen in the extracellular matrix at cell-cell contact, and collagen type 4 (*COL4A1* and *COL4A2*), major collagen for the basement membrane were statistically increased in the IPP (+) group (*COL1A1*, *COL4A1*, *COL4A2*; 0.2-, 0.5-, 0.4-Log2 fold changes, respectively). The RNA expression level of hemidesmosome-related genes, such as collagen type 17 (*COL17A1*) and integrin alpha 6 (*ITGA6*)[26] were found to be higher in the IPP (+) group (1.3- and 1.1-Log2 fold changes, respectively). The gene for laminin alpha 5 (*LAMA5*), a component of laminin 511 that is related to the endothelial cell's basement membrane, was found to be elevated (0.8-Log2 fold changes). The data suggests a relationship between IPP and hemidesmosome maturation, a key factor in cell sheet adhesion, endothelial cell migration, and vasculogenesis. Next, Fig. 5d shows the expression of RNA associated with the HIF family. Among the genes related to the HIF family, the HIF-1α (*HIF1A*) and the HIF-1β gene (*ARNT*) exhibited decreased expression in the IPP (+) group, with -0.5- and -0.2-Log2 fold changes, respectively. No significant differences were observed in the expression levels of genes related to HIF-2α (*EPAS1*) and HIF-3α (*HIF3A*). These results suggest a potential relationship between IPP (+) and the inhibition of the HIF pathway. Finally, angiogenesis-related RNA sequencing results are presented in Fig. 5e. The level of RNA expression associated with angiogenesis was reduced in the IPP (+) group for a number of genes, such as hepatocyte growth factor (*HGF*), *FGF7*, angiopoietin 1 (*ANGPT1*), matrix metallopeptidase 2 (*MMP2*) and insulin-like growth factor 1 (*IGF1*). No changes were observed in genes such as *FGF2*, *VEGFA*, angiopoietin-like 2 (*ANGPTL2*), and *ANG*. On the other hand, the RNA expression level of platelet-derived growth factor (PDGF)-related genes such as *PDGFA* and *PDGFB* was increased in the IPP (+) group (*PDGFA*, *PDGFB*; 0.8- and 1.0-Log2 fold changes, respectively). Additionally, the gene related to placental growth factor (*PGF*) was also slightly increased in the IPP (+) group (0.5-Log2 fold change).

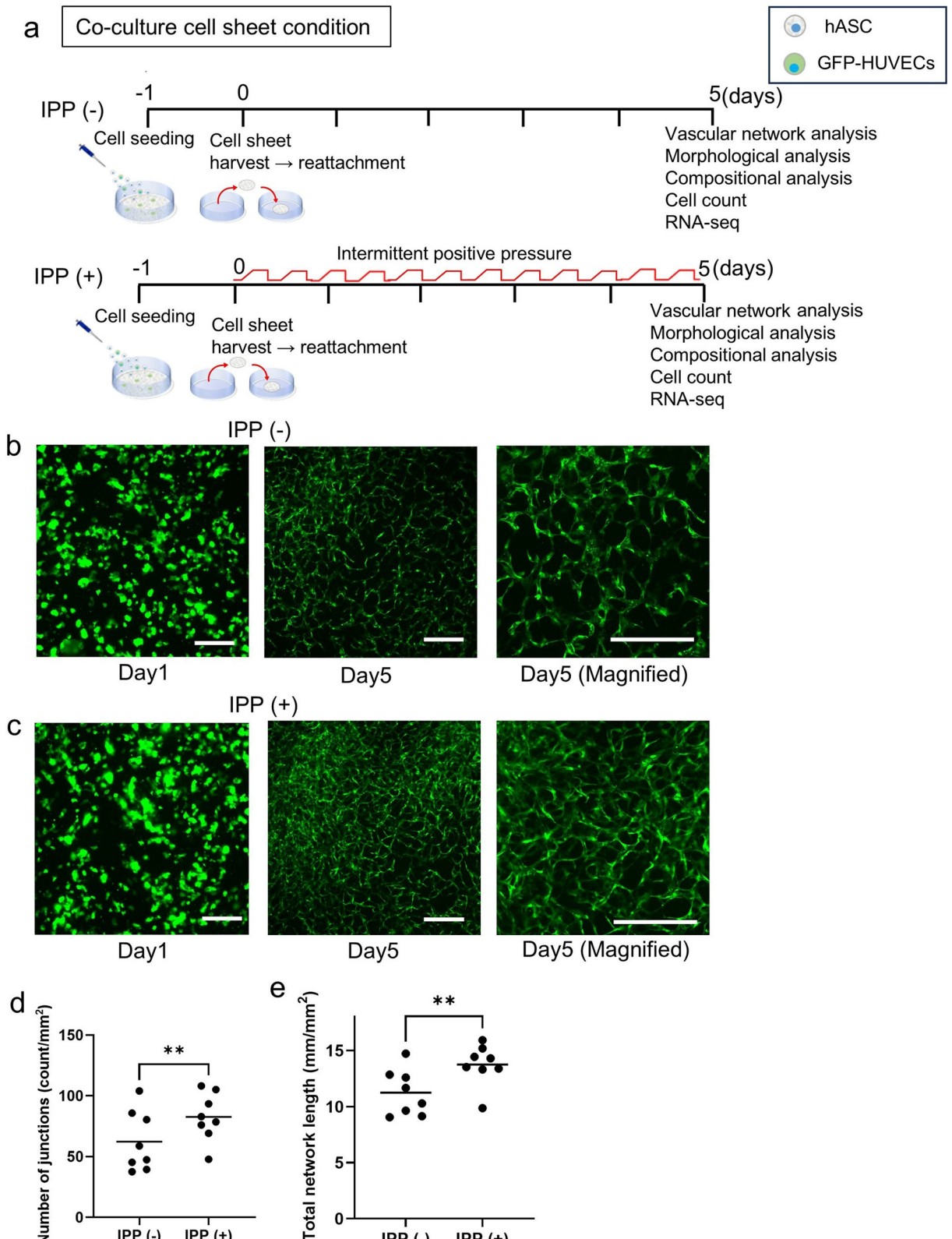

**Fig. 2 | Comparison of vascular endothelial cells network in co-cultured cell sheet condition applied intermittent positive pressure and non-pressurization.**
**a** Diagram of the experiment on the state of the co-cultured cell sheets. According to the schedule shown in the figure, cell seeding, cell sheet production, and evaluation of each item were carried out. **b** Representative images of GFP-HUVECs under IPP (−) on day 1 (*left*) and day 5 (*middle, right*). Scale bar: 500 µm. **c** Representative images of GFP-HUVECs (*green*) under IPP (+) on day 1 (*left*) and day 5 (*middle,*

*right*). Scale bar: 500 µm. **d** Total number of endothelial network junctions on day 5 (*n* = 8, measured from 8 independent culture dishes, with the mean of 3 randomly selected fields of view per dish, \*\**p* < 0.01, Cohen's *d* = 0.91). **e** Total length of the endothelial network on day 5 (*n* = 8, measured from 8 independent culture dishes, with the mean of 3 randomly selected fields of view per dish, \*\**p* < 0.01, Cohen's *d* = 1.29). **d, e** Plots in the graph show the value of each experiment and the bar shows the average.

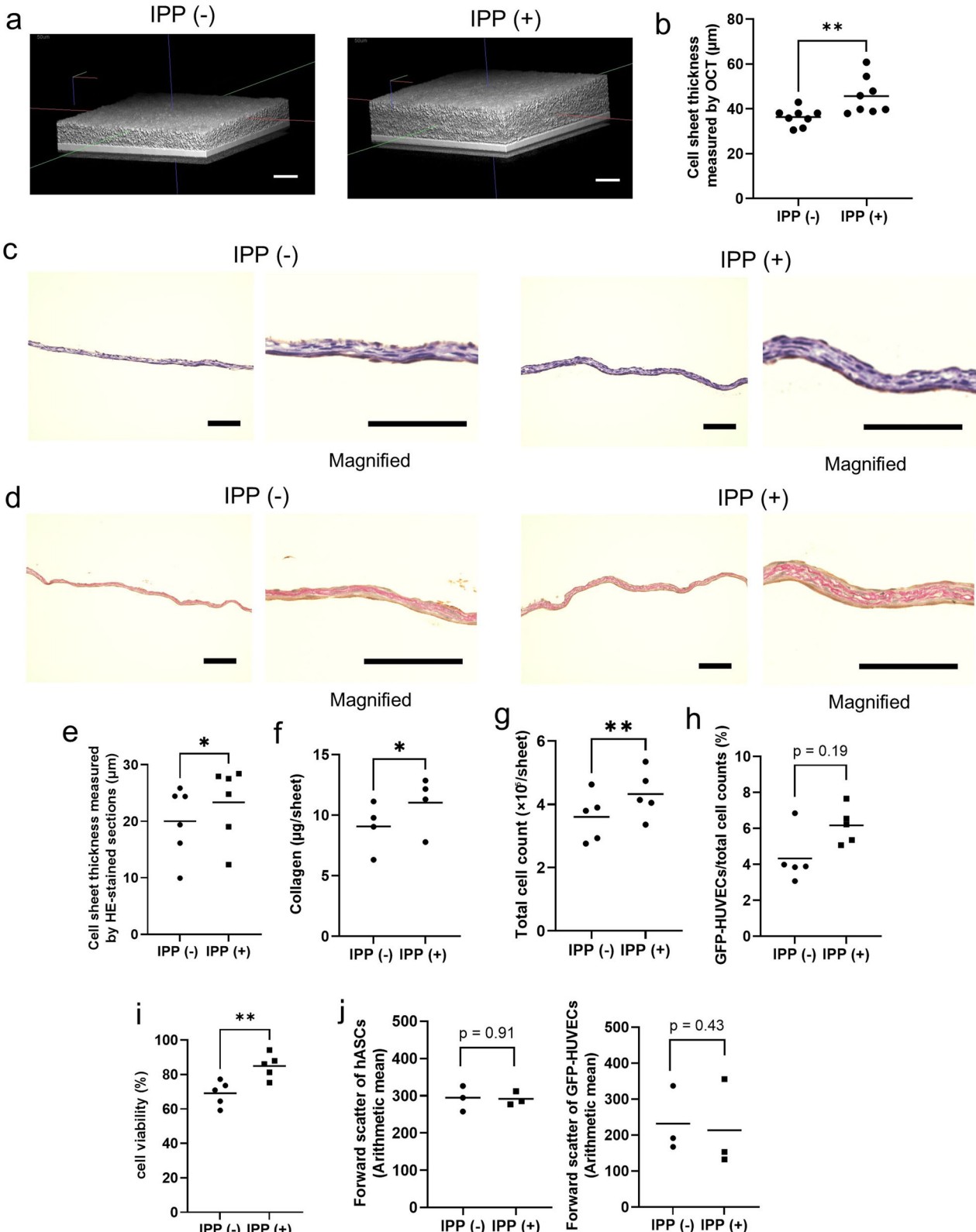

## Intermittent positive pressure does not induce thickening of the co-cultured cell sheet under mild hypoxia

To investigate the effect of increased dissolved oxygen levels induced by IPP, we cultured co-cultured cell sheets under two conditions, IPP (+) and IPP (−), in a mild hypoxia environment (12% $O_2$, 5% $CO_2$), and compared their thickness. Under the mild hypoxia environment, no difference in cell sheet

thickness was observed between the IPP (+) and IPP (−) conditions [IPP (−) vs IPP (+) (µm); 33.4 ± 3.5 vs 33.5 ± 3.3] (Fig. 6a). Furthermore, there was no significant difference when compared to cell sheets cultured under normal conditions (Fig. 3b: IPP (−)), indicating that the thickening effect of IPP on cell sheets is abolished under mild hypoxia conditions. Next, to inhibit eNOS, a phenotype of *NOS3*, which was upregulated by IPP (+) in

**Fig. 3 | Comparison of morphological and compositional changes of co-cultured cell sheets applied intermittent positive pressure and non-pressurization. a** 3D-OCT images of the central section of a co-cultured cell sheet on day 5 days of IPP (−) (*left*) and IPP (+) (*right*). Scale bar: 50 μm. **b** The average thickness of cell sheets on day 5 measured using OCT images (*n* = 8, average thickness measured at 3 random locations on each cell sheet, **$p < 0.01$, Cohen's *d* = 1.43). **c** Representative cross-sectional image of co-cultured cell sheets on day 5 cultured under IPP (−) (*left*) and IPP (+) (*right*) on day 5. Magnified views of the sections are shown alongside. The cell sheets were stained with hematoxylin and eosin (HE-stained). Scale bar: 100 μm. **d** Representative cross-sectional image of co-cultured cell sheets on day 5 cultured under IPP (−) (*left*) and IPP (+) (*right*). Magnified views of the sections are shown alongside. The cell sheet was stained with Sirius red. Scale bar: 100 μm. **e** The

thickness of cell sheets on day 5 measured using HE-stained sections (*n* = 6, average thickness measured at 3 random locations on each cell sheet, *$p < 0.05$, Cohen's *d* = 0.53). **f** The amount of collagen contained in the cell sheet (*n* = 4, independent samples, *$p < 0.05$, Cohen's *d* = 0.91). **g** The total number of the cells in the cell sheet on day 5 (*n* = 8, independent samples, **$p < 0.01$, Cohen's *d* = 0.95). **h** The percentage of GFP-HUVECs of the total cell number in the co-cultured cell sheet (*n* = 5, independent samples, *p* = 0.19, Cohen's *d* = 1.47). **i** The percentage of cell viability in the co-cultured cell sheets on day 5 (*n* = 5, independent samples, **$p < 0.01$, Cohen's *d* = 2.2). **j** The forward scatter of hASCs (*left*) (*n* = 3, independent samples, *p* = 0.91, Cohen's *d* = 0.05) and GFP-HUVECs (*right*) (*n* = 3, independent samples, *p* = 0.43, Cohen's *d* = 0.17) measured using a flow cytometry.

**Fig. 4 | Comparison of air saturation at the bottom of the dish applied intermittent positive pressure and non-pressurization. a** Schematic diagram for the measurement of air saturation at the bottom of the dish in the area surrounding the cell sheet. **b** Representative data showing relative changes in air saturation. **c** Relative air saturation ratio of each experiment measured under IPP (−) / IPP (+) after 12 h, 24 h, and 48 h (*n* = 3).

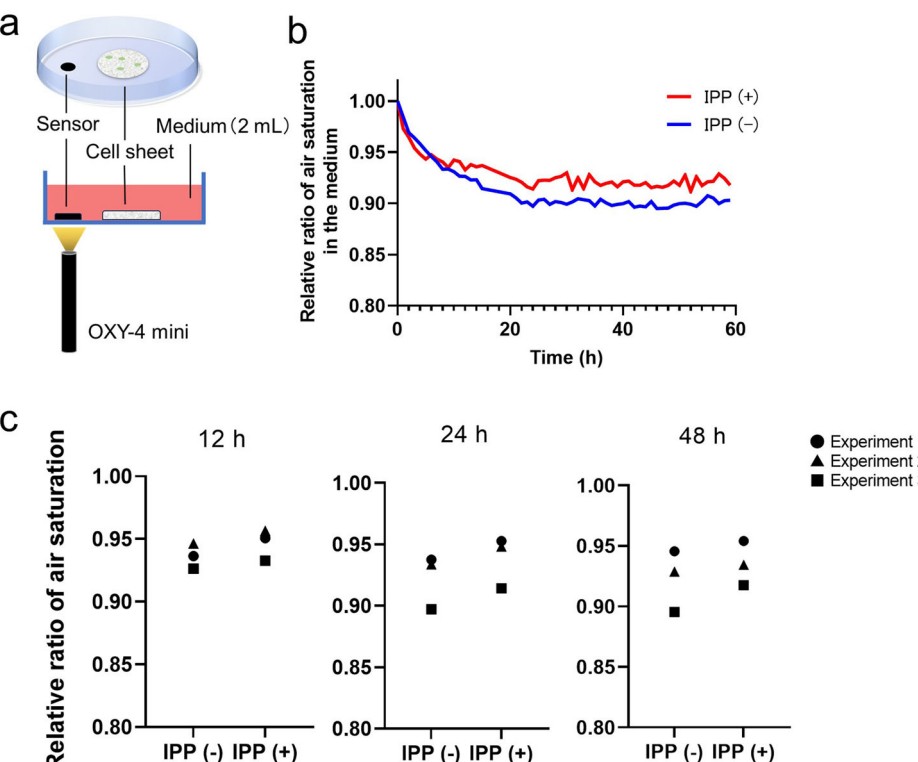

the RNA-seq analysis (Fig. 5a), we added the eNOS inhibitor L-NAME and compared the thickness of co-cultured cell sheets under IPP (+) and IPP (−) conditions. The results showed that the IPP (+) group was significantly thicker than the IPP (−) group [L-NAME (+) IPP (−) vs L-NAME (+) IPP (+) (μm); 35.3 ± 4.9 vs 41.4 ± 6.3] (Fig. 6e). The inhibition of eNOS by L-NAME did not suppress the thickening effect of IPP.

### eNOS inhibition suppresses vascular network formation in co-cultured cell sheets under intermittent positive pressure

To investigate the effect of IPP on vascular endothelial network formation, we added the eNOS inhibitor L-NAME to the co-cultured cell sheets and compared vascular endothelial network formation under IPP (+) and IPP (−) conditions (Fig. 6b, c). Treatment with L-NAME under IPP (−) conditions did not lead to any changes in the formation of the vascular endothelial network compared to the group cultured under IPP (−) conditions without L-NAME [IPP (−) L-NAME (−) vs IPP (−) L-NAME (+); Total number of endothelial network junctions (count/mm²): 60.2 ± 16.6 vs 59.5 ± 14.6, Total length of the endothelial network (mm/mm²): 5.7 ± 0.8 vs 5.7 ± 0.7]. In contrast, when the cell sheets were cultured under IPP (+) conditions, the addition of L-NAME resulted in a suppression of vascular network formation compared to the group without L-NAME [IPP (+) L-NAME (−) vs IPP (+) L-NAME (+); Total number of endothelial network junctions (count/mm²): 84.1 ± 11.7 vs 67.4 ± 17.0, Total length of the

endothelial network (mm/mm²): 6.7 ± 0.5 vs 6.1 ± 0.6]. Furthermore, no significant differences in vascular network formation were observed when comparing the L-NAME-treated cell sheets cultured under IPP (−) and IPP (+) conditions [IPP (−) L-NAME (+) vs IPP (+) L-NAME (+); Total number of endothelial network junctions (count/mm²): 59.5 ± 14.6 vs 67.4 ± 17.0, Total length of the endothelial network (mm/mm²): 5.7 ± 0.7 vs 6.1 ± 0.6] (Fig. 6b, c). Inhibition of eNOS by L-NAME suppressed the vascular network formation induced by IPP (+), indicating the involvement of eNOS in the process.

### Evaluation of the impact of intermittent positive pressure on the thickness and functionality of co-cultured cell sheets transplanted in vivo

To evaluate the functional impact of IPP on co-cultured cell sheets in vivo, the cell sheets in the IPP (−) and IPP (+) groups were detached from the temperature-responsive dish on day 5 (Fig. 7b, e) and transplanted on a superficial gluteal muscle of a rat. The cell sheet was constructed using GLuc-hASCs, which are hASCs transduced with lentiviral vectors encoding the Gaussian luciferase gene and the m-Scarlet-I gene, and GFP-HUVECs were co-cultured. The evaluation of GLuc level in rat blood and blood perfusion ratio in the cell sheet was conducted 2 days later (Fig. 7a). Both groups of the transplanted cell sheet were successfully engrafted (Fig. 7c, f). The thickness of the cell sheets was found to be significantly thicker in the

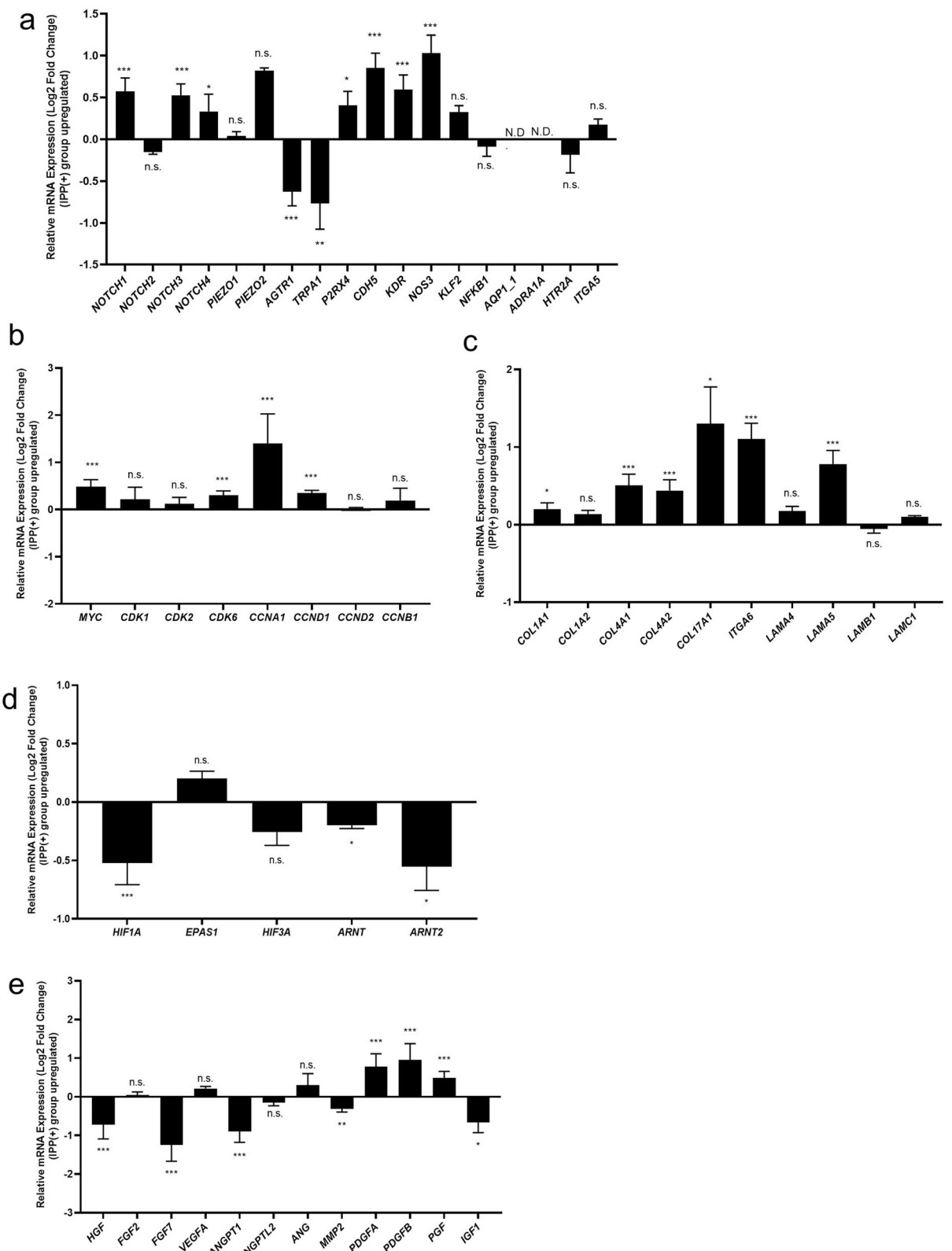

**Fig. 5 | Comparison of RNA expression in co-cultured cell sheets between intermittent positive pressure and non-pressurization. a** The relative expression of RNAs associated with endothelial mechanotransduction induced by shear stress ($n = 3$, independent samples). **b** The relative expression of RNAs associated with cell cycles ($n = 3$, independent samples). **c** The relative expression of RNAs associated with the extracellular matrix (ECM) and hemidesmosome ($n = 3$, independent samples). **d** The relative expression of RNAs associated with hypoxia-inducible factor (HIF) family ($n = 3$, independent samples). **e** The relative expression of RNAs associated with angiogenesis factors ($n = 3$, independent samples). **a–e** The comparison was made between co-cultured cell sheets cultured under IPP (−)/IPP (+). The cell sheets were cultured for 5 days.

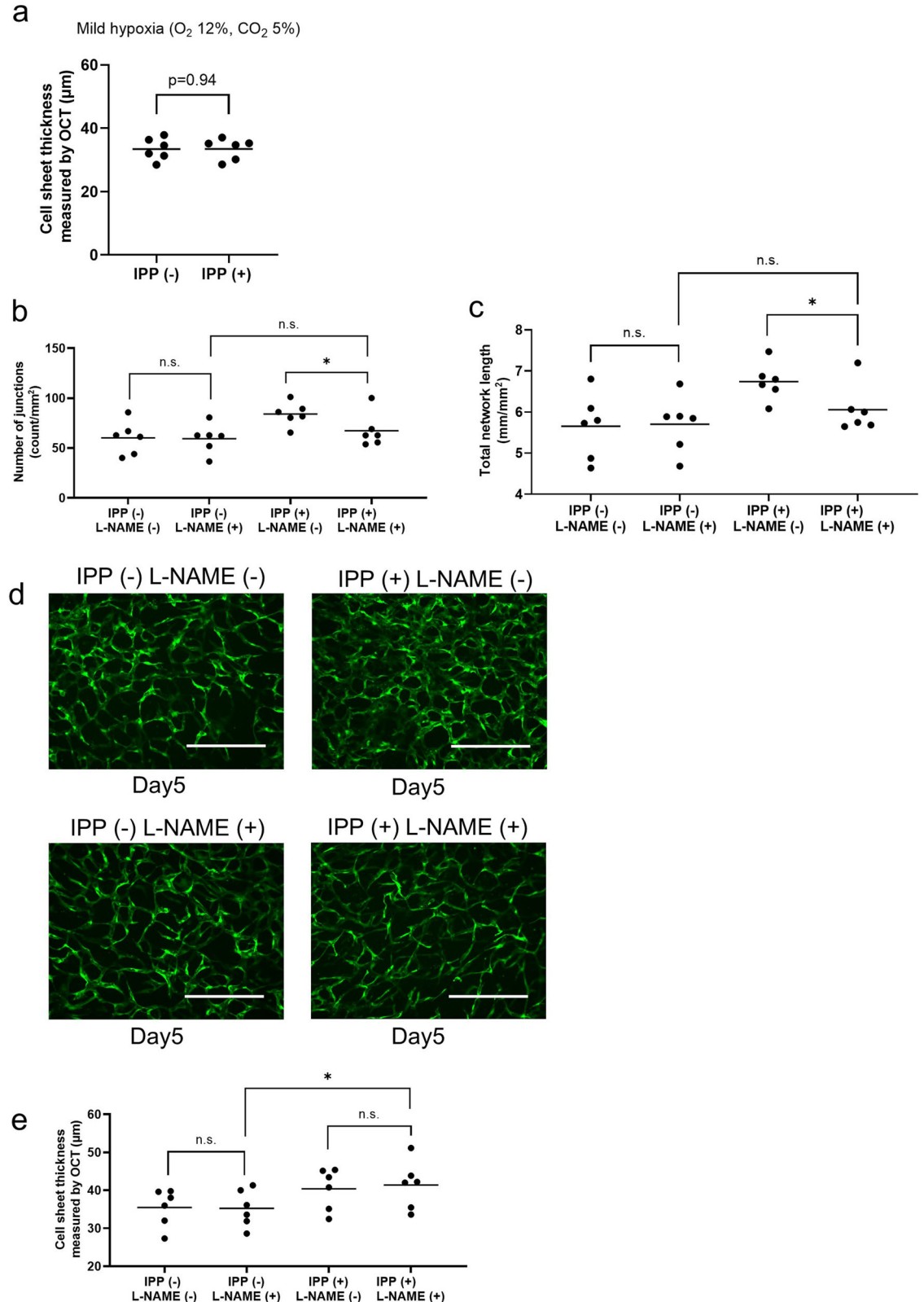

IPP (+) group [IPP (−) vs IPP (+) (μm); 21.9 ± 5.6 vs 27.1 ± 4.2] (Fig. 7h, i). The intensity of GLuc luminescence in rat blood tends to be higher in the IPP (+) group [IPP (−) vs IPP (+) (AU); 2600 ± 224 vs 2824 ± 241, $p = 0.13$, Cohen's $d = 0.96$]: Although the difference did not reach statistical significance ($p = 0.13$), the large effect size in the animal experience (Cohen's $d = 0.96$) suggests a notable trend (Fig. 7j). In the engrafted cell sheet, GFP-

HUVEC networks were observed to form lumen and blood perfusion was observed (Fig. 7k), indicating that the vECs of the donor and host were connected after transplantation. On the other hand, the blood perfusion ratio was not significantly different between the two conditions [IPP (−) vs IPP (+) (perfusion ratio); 1.2 ± 0.2 vs 1.4 ± 0.3] (Fig. 7l). In order to minimize the impact of individual differences between rats on the experimental

**Fig. 6 | Effects of mild hypoxia on intermittent positive pressure-induced responses in co-cultured cell sheets and inhibition of shear stress pathways by L-NAME. a** The average thickness of cell sheets under mild hypoxia on day 5 measured using OCT images [IPP (−) vs IPP (+) (μm): $n = 6$, $p = 0.94$, Cohen's $d = 0.02$]. **b** Total number of endothelial network junctions on day 5 [IPP (−) L-NAME (−) vs IPP (−) L-NAME (+) (count/mm²): $n = 6$, $p = 0.56$, Cohen's $d = 0.05$], [IPP (−) L-NAME (+) vs IPP (+) L-NAME (+) (count/mm²): $n = 6$, $p = 0.44$, Cohen's $d = 0.50$], [IPP (+) L-NAME (−) vs IPP (+) L-NAME (+) (count/mm²): $n = 6$, *$p < 0.05$, Cohen's $d = 1.14$]. **c** The total length of the endothelial network on day 5 [IPP (−) L-NAME (−) vs IPP (−) L-NAME (+) (mm/mm²): $n = 6$, $p = 0.84$, Cohen's $d = 0.06$], [IPP (−) L-NAME (+) vs IPP (+) L-NAME (+) (mm/mm²): $n = 6$, $p = 0.31$, Cohen's $d = 0.56$], [IPP (+) L-NAME (−) vs IPP (+) L-NAME

(+) (mm/mm²): $n = 6$, *$p < 0.05$, Cohen's $d = 1.31$]. **d** Representative image of GFP-HUVECs (*green*) of each group on day 5. Scale bar: 500 μm. **e** The average thickness of cell sheets on day 5 measured using OCT images of each group [IPP (−) L-NAME (−) vs IPP −) L-NAME (+) (μm); 35.5 ± 4.9 vs 35.3 ± 4.9, $n = 6$, $p = 0.91$, Cohen's $d = 0.04$], [IPP (−) L-NAME (+) vs IPP (+) L-NAME (+) (μm); 35.3 ± 4.9 vs 41.4 ± 6.3, $n = 6$, *$p < 0.05$, Cohen's $d = 1.09$], [IPP (+) L-NAME (−) vs IPP (+) L-NAME (+) (μm); 40.4 ± 5.4 vs 41.4 ± 6.3, n = 6, $p = 0.58$, Cohen's $d = 0.17$]. **a**, **e** $N = 6$, representing 6 cell sheets, with the average thickness measured at 3 random locations on each sheet. **b**, **c** $N = 6$, representing 6 cell sheets, with the evaluation of the endothelial vascular network performed in 3 randomly selected fields of view per sheet.

outcome, the co-cultured cell sheets, which were cultured under IPP (−) or IPP (+) conditions, were transplanted into the right and left superficial gluteal muscles of the same rat. Subsequently, the blood perfusion ratio was again compared. There was no statistically significant difference in blood perfusion ratio between the IPP (−) and IPP (+) conditions [IPP (−) vs IPP (+) perfusion ratio: 1.2 ± 0.1 vs. 1.2 ± 0.1, $n = 7$, $p = 0.35$, Cohen's $d = 0.60$] (Supplementary Fig. 4g), which is similar to the results observed in Fig. 7l.

### Islet beta cells in a tri-cultured condition demonstrated better glucose-responsive insulin secretion when cultured under intermittent positive pressure

The results shown in Fig. 7j suggest that IPP culture is an appropriate approach for endocrine cells. With a view to the therapeutic application of culturing under intermittent positive pressure, iGL (a clonal INS-1E cell line derived from rat pancreatic β-cells and stably expressing a fusion protein of insulin and Gaussia luciferase (insulin-GLuc)), hASCs and GFP-HUVECs were tri-cultured (Fig. 8b). The glucose-responsive insulin secretion test was performed on day 3 (Fig. 8a). The results showed that after stimulation with 2 mM glucose, the insulin-GLuc luminescence intensity was 196.5 ± 59.5 (AU) in the IPP (−) group and 165.8 ± 66.9 (AU) in the IPP (+) group ($n = 4$). After stimulation with 20 mM glucose, the insulin-GLuc luminescence intensity was 194.0 ± 25.4 (AU) in the IPP (−) group and 210.3 ± 52.7 (AU) in the IPP (+) group ($n = 4$). The glucose-stimulated insulin secretion index was statistically higher in the IPP (+) group (1.3 ± 0.2) compared to the IPP (−) group (1.0 ± 0.2) (Fig. 8c). Thus, IPP culture would be a more attractive method for endocrine tissue engineering in the medical treatment of endocrine disorders.

## Discussion

The present study has identified the following three major findings: (i) IPP, which was generated by an air-compressing pressurization device, promoted vEC network formation in the co-culture cell sheet of MSCs and vECs. (ii) The IPP culture method demonstrated an ability to promote aerobic metabolism in cells, while also favoring the secretion of pro-angiogenic factors and the synthesis of ECM. This culture method facilitates the growth of the co-cultured cell sheets into a thick tissue sheet that is rich in vascular endothelial networks. (iii) The transplantation of a cell sheet cultured under IPP demonstrated successful engraftment while maintaining its thickness. The results also suggested a potential improvement in certain protein secretion into the systemic circulation.

Previous studies of pressure stimulation on vECs have used a variety of pressure stimulation methods. The pressure method reported by Yoshino et al. involves sandwiching HUVECs with collagen gel, compressing the HUVECs and the medium together with a device, and applying a pressure of 50 mmHg for 3 h[13]. Alternatively, the pressure method reported by Sekiya et al. involves increasing the depth of the medium (90 mm)[17]. In a study by Morrow et al. HUVECs were seeded onto a soft material culture dish, and mechanical stimulation was applied to the cells by deforming the bottom of the dish using vacuum pressure for a 24-h period[16]. In these studies, all pressurization stimulation was found to promote vascular networking. The IPP method in the co-culture environment of the present study also demonstrated a similar effect in promoting vascular network formation

within a cell sheet structure, despite clear differences in the pressure application methods. However, from the standpoint of gene expression, the IPP method demonstrated different results compared to the other reported studies. The present study demonstrated elevated expression of VE-cadherin in the IPP (+) group. VE-cadherin has been reported to be involved in vascular tube formation[27], and is thought to be involved in the mechanism of vascular network construction by IPP. Conversely, Yoshino et al. reported that pressurization stimulation did not affect the expression of VE-cadherin[13], which differs from our results in the present study. The observed difference in VE-cadherin expression could be related to the difference in mechanical stimulation. Increased expression of VE-cadherin has been reported as a factor related to shear stress[28] among the many possible mechanical stimuli. Conversely, with regard to hydrostatic pressure, although the mechanism is largely unknown, there are reports that VE-cadherin is down-regulated by hydrostatic pressure[29]. In the present experiment, the expression of VE-cadherin increased, as did the expression of multiple related genes downstream of VE-cadherin signals (VEGFR2 gene *KDR*, eNOS gene *NOS3*, and cyclin A gene *CCNA1*) caused by shear stress[15]. Furthermore, eNOS inhibition experiments in the present study demonstrated that inhibiting eNOS under IPP (+) conditions significantly suppressed vascular network formation, bringing it to levels comparable to those observed under IPP (−) conditions. This result is consistent with the RNA-seq findings and suggests a strong correlation between vascular network formation induced by the IPP method and shear stress. In contrast, the increased expression of bFGF[30], TRPV4[31], integrin α5[32], AQP1, α1-AR, and SR-2A[13], which have all been reported to be associated with hydrostatic stimulation, was not confirmed in the present study. Thus, IPP-induced pressurization promoted the vascular endothelial network mainly through a mechanical signal similar to shear stress, rather than a mechanical signal from hydrostatic pressure. The differences in signal pathways observed in the gene expression analysis can likely be attributed to differences in the "intermittent" and "continuous" pressurization methods. Therefore, further research is warranted to elucidate the relationship between pressurization methods and mechanical signaling pathways.

It has been reported that angiogenesis changes in vascular endothelial cells due to hypoxia are observed at oxygen concentrations of 1–10%[33,34]. In contrast to our system, although the bottom of the culture dish became a more aerobic environment, it succeeded in promoting a vascular endothelial network. In fact, *HIF1A* induced by hypoxia was lower in cell sheets in the IPP (+) group compared to the IPP (−) group. It has been reported that numerous pro-angiogenic factors are upregulated by *HIF1A*[35]; however, in the present experiment, *HIF1A* exhibited a decrease in the IPP (+) group, accompanied by a reduction in the expression of numerous HIF1A-related angiogenesis-promoting factors such as *ANGPT1*, *FGF7*, *MMP2*. Conversely, the expression of *PDGFA*, *PDGFB*, and *PGF* increased in the IPP (+) group. *PDGFB* and *PGF* are factors that are related to *HIF1A*[36], but their expression is also enhanced by mechanical stimulation[37,38]. We believe that the promotion of a vascular endothelial network by IPP is not mediated by *HIF1A*, but is mainly due to the angiogenesis pathway induced by shear stress.

One of the most notable findings of the present study is the thickening of the cell sheet under IPP. Kojima et al. have demonstrated that mechanical stimulation can result in increased thickness of cell sheets. They also

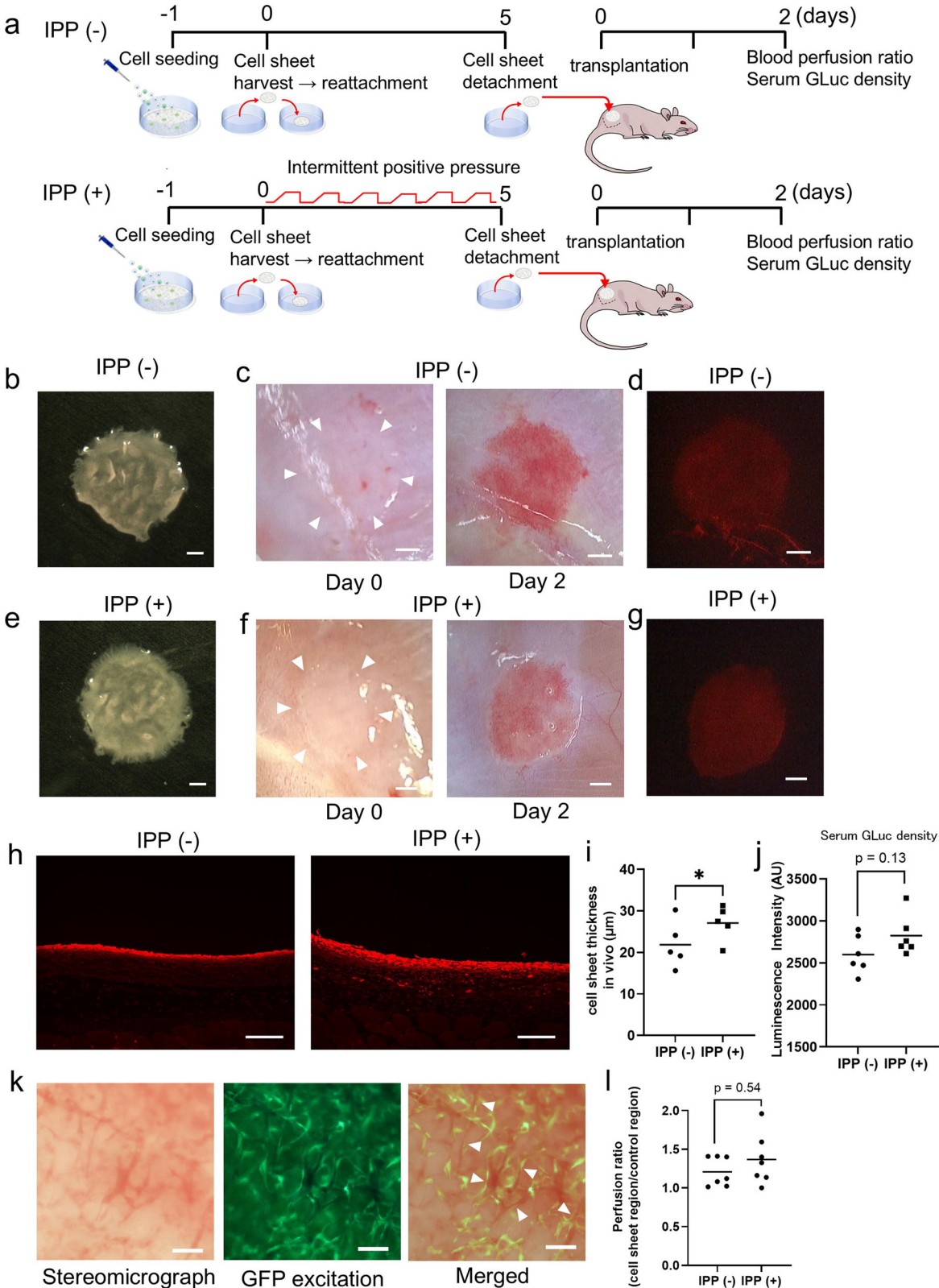

observed that when a human umbilical artery smooth muscle cell sheet, which was harvested using a fibronectin-coated atelocollagen membrane, was cultured under mechanical stimulation, there was an increase in collagen RNA expression and an enhancement in cell sheet thickness[39]. In both Kojima et al. and our experimental systems, collagen RNA expression levels were increased in pressurized cell sheets. In pressurized cell sheets, Kojima

et al. observed increased expression of lysyl oxidase (*LOX*), an enzyme involved in the strength of extracellular matrices that is essential for the early steps of cross-linking of extracellular matrices such as collagen fibrils. Conversely, there was no discernible difference in *LOX* levels in our system. It is important to note that the oxygen conditions in our experimental systems differed. While Kojima et al. used pressurization in a hypoxic

**Fig. 7 | Functional comparison of intermittently positive pressurized and non-pressurized cell sheets transplanted subcutaneously in rats. a** Diagram of the experiment on the transplantation of co-cultured cell sheets. **b, e** Representative stereomicroscope images of a co-cultured cell sheet. The cell sheets were detached from the dish on day 5. Scale bar: 1 mm. **c, f** Representative images of a co-cultured cell sheet cultured under just after transplantation (*left*) and 2 days after transplantation (*right*). The white arrowheads indicate the region of the engrafted cell sheet. Scale bar: 1 mm. **d, g** Representative fluorescence image of a cell sheet (**c** or **f**) on day 2. The cell sheets are indicated in red as a consequence of m-Scarlet-I. Scale bar: 1 mm. **h** Representative images of fluorescence immunostained sections of the engrafted cell sheet 2 days after transplantation, IPP (−) (*left*) and IPP (+) (*right*). The cell sheets are labeled in red with the anti-RFP antibody. Scale bar: 100 μm.

**i** Thickness of the engrafted cell sheets. (*n* = 5, average thickness measured at 3 random locations on each cell sheet, *$p < 0.05$, Cohen's $d = 1.06$). **j** The intensity of Gaussia luciferase (GLuc) luminescence in blood serum on day 2 following cell sheet transplantation. (*n* = 6, independent samples, $p = 0.13$, Cohen's $d = 0.96$).
**k** Representative image of a transplanted co-cultured cell sheet captured by a stereomicroscope (*left*) and fluorescence microscope (*middle*). The merged image is shown on the right. The engrafted GFP-HUVECs formed tubular structures, which are indicated by the white arrows. It appears that the tube was filled with blood from the recipient rat. Scale bar: 100 μm. **l** Perfusion ratio of transplanted cell sheets on day 2 following transplantation. (*n* = 7, independent samples, $p = 0.54$, Cohen's $d = 0.60$). GLuc: Gaussia luciferase.

**Fig. 8 | Comparison of the ability of glucose-responsive insulin secretion of iGL cells tri-cultured in planar condition under intermittent positive pressure and non-pressurization.**
**a** Diagram of tri-culture planar condition experiment including iGLs. **b** Representative image of tri-cultured cells under IPP (−; left) and IPP (+; right). The red color indicates iGLs, the green color indicates GFP-HUVECs and the blue color indicates the nucleus. Scale bar: 100 μm. **c** Glucose-stimulated insulin secretion index of iGLs cultured in planar condition under IPP (−) / IPP (+) on day 3. (*n* = 4, independent samples, *$p < 0.05$, Cohen's $d = 1.36$).

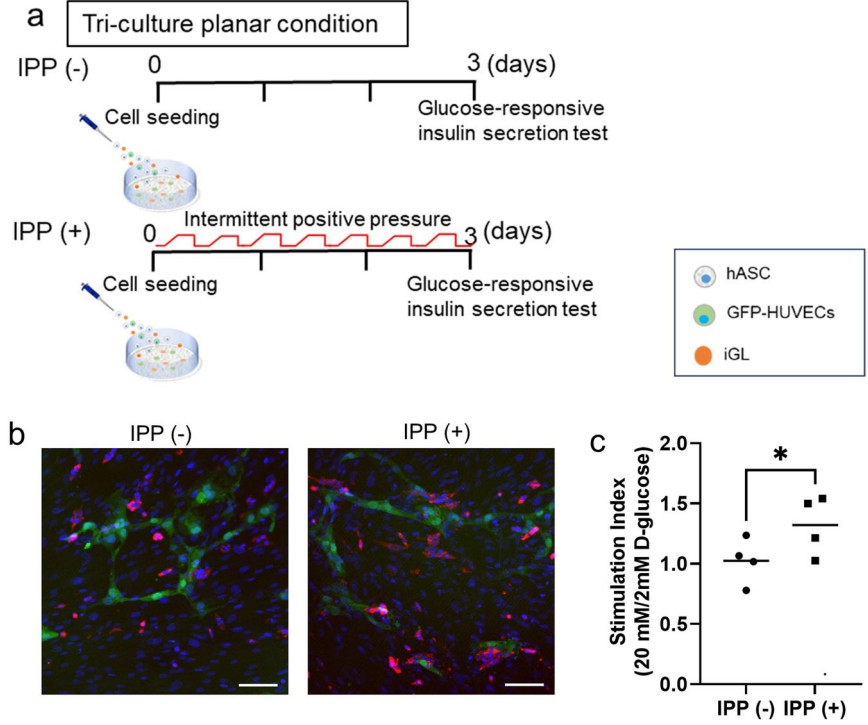

environment, in our system the dissolved oxygen in the medium increased due to pressurization. In the present study, under the mild hypoxia environment, IPP did not induce thickening of the co-cultured cell sheet. However, even when eNOS, one of the shear stress pathways induced by IPP, was inhibited, the cell sheet still thickened. These findings suggest that while the shear stress pathways stimulated by IPP are not limited to eNOS, shear stress alone is insufficient to induce cell sheet thickening. Instead, the increase in dissolved oxygen levels appears to play a predominant role in this process. Oxygen plays a significant role in procollagen polymerization into collagen. Proline residue hydroxylation, catalyzed by the P4H family, is crucial in the collagen polymerization process, and P4H-catalyzed hydroxylation reactions require oxygen[40]. Therefore, the increase in dissolved oxygen in our experimental system may have influenced the increase in collagen content within the cell sheet. Additionally, in the present study, the number of cells under IPP culture was found to be greater than under non-IPP culture. RNA-seq analysis revealed the upregulation of *NOTCH1*, *NOTCH3*, and *NOTCH4* signaling and their downstream targets, including *MYC*, *CCND1*, *CDK6*, and *CCNA1*, in the IPP (+) group. These genes, particularly *MYC*, are known to be associated with the cell cycle progression of MSCs[25]. Furthermore, *NOTCH1* is recognized as a key component of the shear stress signaling pathway. These findings are consistent with the hypothesis that the activation of shear stress signals induced by IPP promotes cell proliferation. While the thickening of the cell sheet under IPP conditions is primarily attributed to ECM components such as collagen, the

difference in cell numbers also appears to contribute to this outcome. Further investigations are required to clarify the effects of IPP-induced NOTCH signaling activation on cells.

In the present study, cell sheets cultured under IPP and normal culture conditions were subcutaneously transplanted into rats. Previous reports have indicated that the construction of a vascular endothelial network within the tissue is associated with enhanced cell sheet engraftment and the formation of vascular lumen structures[41]. However, in the present study, no difference was observed in the blood perfusion ratio within the cell sheet after engraftment. In this experiment, although the cell sheets in the IPP (+) group exhibited a greater density of vascular endothelial networks, both groups demonstrated the formation of a vascular endothelial network. Consequently, it is possible that there was little difference in the blood perfusion ratio, which is an indirect indicator of vascularization, or that the size of the graft used in our study was too small to demonstrate a difference. Conversely, the thicker cell sheet cultured in an IPP environment retained this thickness following transplantation, in comparison to normal culture. Furthermore, GLuc secretion from transplanted hASCs was observed to be higher in the IPP environmental cell sheet, which may reflect the fact that the number of hASC cells was higher in the IPP (+) group. These two results indicate that the transplanted cell sheets cultured under IPP were firmly engrafted. Then, the results of the transplantation of cell sheets cultured under IPP suggest that the serum concentrations of certain secreted proteins also increase. However, to firmly demonstrate that the secretion of other

functional proteins is also increased, further studies, such as using genetic engineering to attach markers to the secreted proteins of transplanted cells, would be needed.

In the present study, we demonstrate the applicability of our method to the field of regenerative medicine. In our experiments, we compared IPP (+) and IPP (−) under three co-culture conditions containing pancreatic islet β cells, which we have previously reported as pancreatic islet β cell tissues[4,8]. Our results indicate that the insulin secretion ability of pancreatic islet β cells was significantly higher in IPP (+) than in IPP (−) under all three co-culture conditions. It is well established that pancreatic islet β cells are sensitive to hypoxic environments, where improved oxygenation can enhance glucose-responsive insulin secretion[42]. Improving oxygenation in pancreatic islet β cell co-culture using IPP is an effective approach. We have shown that the IPP culture method simultaneously achieves aerobic metabolism and vascular endothelial network formation in a co-culture system. Since the IPP method enables the construction of vascularized tissue containing a greater number of cells and ECM, it may be applicable to the production of transplants for cells with a higher oxygen demand, such as pancreatic islet β cells.

The vascular endothelial network is influenced by oxygen concentration, and in our pressurization system, the balance between mechanical stimulation and oxygenation may have impacted the outcome. Therefore, it is essential to optimize pressure conditions according to the cells co-cultured with blood vessels. In other words, it is necessary to increase oxygenation for cells with high oxygen demand, such as pancreatic islet β cells, and for cell sheets seeded at a higher density than in the present study.

In conclusion, IPP has the potential to be employed to construct three-dimensional tissues well-developed vascular networks. Moreover, the capacity to enhance the secretion function of cells with endocrine capabilities is also proposed. Consequently, by applying this method to various endocrine cells, it can be utilized in the fabrication of transplantable three-dimensional tissues with enhanced endocrine functions.

## Materials and methods
### Ethic statement
All animal experiments were performed following the guidelines of the Ethics Committee for Animal Experimentation of Tokyo Women's Medical University and in compliance with the Legislation and Regulation on the use of animals in biological research with the ARRIVE guidelines. All experimental protocols were approved by the Ethics Committee for Animal Experimentation of Tokyo Women's Medical University (Approval Number AE23-122, AE24-021). All animals were housed in individual cages with free access to food and water under a light/dark cycle of 12 h and maintained at constant room temperature and humidity. Animals were euthanized by exsanguination under 5% isoflurane in accordance with the American Veterinary Medical Association (AVMA) euthanasia guidelines.

### Culture of cells
The GFP-HUVECs were provided by Angio-Proteomie (Boston, MA, USA) and cultured in KBM VEC-1 medium (Kohjin Bio, Saitama, Japan). The hASCs were provided by Lonza Japan (Tokyo, Japan) and cultured in KBM ADSC-1 medium (Kohjin Bio, Saitama, Japan). In the in vivo experiment shown in Fig. 7, GLuc protein and m-Scarlet-I expressing-hASCs (GLuc-hASCs) were used. The Gluc-hASCs were transduced with lentivirus encoding both the GLuc and m-Scarlet-I genes. The iGL cells, a clonal INS-1E cell line derived from rat pancreatic β-cells and stably expressing a fusion protein of insulin-GLuc, were provided by Cosmo Bio Co., Ltd (Tokyo, Japan). The cells were cultured in an IGLM medium (Cosmo Bio Co., Ltd). For this experiment, cells of all types were used at passage p6–9.

### Preparation of co-cultured cells in a planar environment
For Fig. 1a experiments, the planar co-cultured cells were prepared as below. The 35 mm dish was coated with FBS overnight, and GFP-HUVECs and hASCs were seeded in a ratio of 4:1 with a total of $1.0 \times 10^6$ cells. The cells were cultured in the basal medium as described below: DMEM/F-12, GlutaMAX™ supplement medium (Thermo Fisher Scientific, Waltham

MA, USA) containing 10% fetal bovine serum (FBS) and penicillin-streptomycin solution (×100) (FUJIFILM Wako Pure Chemical Corporation, Osaka, Japan).

### Preparation of co-cultured cell sheets for in vitro experiments
For experiments in vitro, two different sizes of co-cultured cell sheet were prepared with hASC and GFP-HUVECs at a ratio of 4:1. One (small-sized cell sheet) is prepared by seeding a total of $5.0 \times 10^5$ cells into a 12-well temperature-responsive plate (UpCell; CellSeed, Inc., Tokyo, Japan), while the other (standard-sized cell sheet) is prepared by seeding a total of $1.0 \times 10^6$ cells into a 35 mm temperature-responsive dish (UpCell; CellSeed, Inc.). In the in vitro experiments, standard-sized cell sheets were used only for the measurement of dissolved oxygen in the medium (Fig. 4), and for the comparison of vascular formation between the planar co-cultured condition and co-cultured cell sheet condition (Supplemental Fig. 2d, e) while small-sized cell sheets were used for the other experiments. Before cell seeding, each temperature-responsive dish was coated with FBS overnight. The cells were cultured in a basal medium with 40 μg/mL of L-ascorbic acid phosphate magnesium salt n-hydrate (ASA) (FUJIFILM Wako Pure Chemical Corporation). ASA was included because of its ability to promote collagen secretion from MSCs, which was expected to enhance the formation of well-structured cell sheets[43]. After 24 h of incubation, the cells with the dish were transferred to an incubator set at 20 °C for 30 min to harvest them as a cell sheet. Finally, the cell sheets were reattached to a temperature-responsive dish for collagen quantification and to a 35 mm cell culture dish for the other experiments.

### Preparation of co-cultured cell sheets for in vivo experiment
The cell sheets were prepared with GFP-HUVECs and hASCs or GFP-HUVECs and GLuc-hASCs at a ratio of 4:1, with a total $1.0 \times 10^6$ cells, following the same procedures as co-cultured cell sheets for the in vitro experiments. The cell sheets were reattached to a temperature-responsive 35 mm culture dish and cultured for 5 days under IPP (+) or IPP (−) conditions (Fig. 7a, Supplementary Fig. 4).

### Preparation of tri-cultured cells in planar condition for insulin-GLuc secretion test
iGL cells, GFP-HUVECs, and hASCs were co-cultured at a ratio of 1:1:4, with a total of $1.2 \times 10^6$ cells per well. The cells were seeded onto FBS-coated 35 mm dishes and cultured in basal medium for 3 days (Fig. 8a).

### Pressurization system
IPP was applied using a pressurizing bioreactor system: A pressure Stimulation Unit connected to a pressurizing chamber (Tokaihit Co., Ltd., Shizuoka, Japan) (Fig. 9a). The pressurization unit and the pressure chamber were placed in an incubator set at 5% $CO_2$ and 37 °C. In the present study, an external pressure of 80 mmHg was intermittently applied to the chamber for 120 s and then 60 s rest. The pressure gradually increased during the 120 s pressurizing phase, taking approximately 60 s to reach 80 mmHg and then held for a further 60 s.

### Imaging devices
An image of a co-cultured cell sheet presented in Supplementary Fig. 2a was captured using an iPhone 12 mini (Apple Inc., Cupertino, CA, USA). Images of co-cultured cells in a planar environment (Fig. 1b, c) were captured using an Eclipse Ts2-FL fluorescence microscope (NIKON CORPORATION, Tokyo, Japan). Images of GFP-HUVECs in the planar and cell sheet environments (Supplementary Fig. 2c) and co-cultured cell sheets (Figs. 2b, c and 6d), and tissue sections (Figs. 3c, d and 7h) were captured using a fluorescence microscopy system (BZ-X800; Keyence Corporation, Osaka, Japan). The morphology of the cell sheets was observed using 3D-OCT (Cell3iMager Estier; SCREEN Holdings Co., Ltd., Kyoto, Japan) (Fig. 3a). Images to analyze the thickness of the cell sheets attached to the dishes were acquired using optical coherence tomography (OCT; Santec Holdings Co., Aichi, Japan) (Figs. 3b and 6a, e). Images shown in Fig. 7b, e. k

**Fig. 9 | A pressurization bioreactor system used in the experiment. a** A pressurization bioreactor system, consists of a control unit, a pressurizing pump unit, and a custom-made pressure chamber. Air is introduced into the chamber through the pump's rotation via the tube. The volume of the chamber is calculated to be 803,840 mm³, based on its dimensions of 80 mm × 80 mm × 3.14 × 40 mm. A pressure of +80 mmHg was applied to the chamber every 180 s. IPP: Intermittent positive pressure.

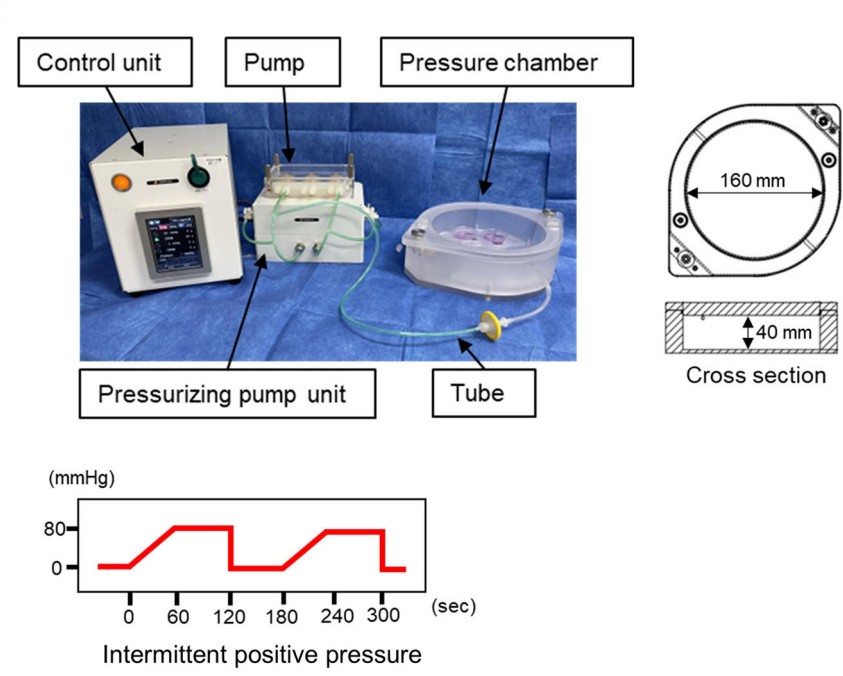

and Supplementary Fig. 4a–d were captured using a stereomicroscope with an image capture system (MVX10 and cellSens Dimension, Olympus, Tokyo, Japan). Images of the cell sheet after transplantation shown in Fig. 7c, d, f, g and Supplementary Fig. 4e were captured using a fluorescence stereomicroscopy system (M651, Leica Microsystems, Wetzlar, Germany; 3CCD camera, Toshiba Corp., Tokyo, Japan; and Video Capture Box, I-O Data Device, Kanazawa, Japan). Images of the tri-cultured cells in Fig. 8b were captured using a confocal laser scanning microscopy system (Laser Scanning Microscope FV1200-IX83, Olympus).

### Evaluation of vascular endothelial network of GFP-HUVECs

The total number of junctions and total network length were analyzed using Angio Tool v0.5a software (National Institute of Health, Maryland, USA). The average number of junctions and total network length of 3 randomly selected fields from each section were used for analysis, where $n = 1$.

### Evaluation of the morphology of the cell sheet attached to the dishes and analysis of the thickness

The cross-sectional area and length of the cell sheet in the OCT image were measured with ImageJ 1.53q software (National Institute of Health). For the analysis, the average thickness of three random sections of each cell sheet was used. The cross-sectional area of the HE-stained cell sheet was measured using the Hybrid Cell Count Application (BZ-H4C) analysis software (Keyence Corporation). The length of cell sheets was measured using ImageJ 1.53q software (National Institute of Health).

### Measurement of the ratio of lactate production to glucose consumption (L/G ratio)

The concentration of glucose and lactate in the media of cells in the planar condition was measured using Stat strip Express 900 (Nova Biomedical, Waltham, USA) and Stat strip Lactate (Nova Biomedical, Waltham, USA), respectively (Fig. 1a, f). To calculate the L/G ratio, lactate production values (mmol⁻¹) were divided by glucose consumption values (mmol⁻¹).

### Cell counts of co-cultured cells in planar and cell sheet environment

Co-cultured cells in the planar environment were treated with Accutase (Innovative Cell Technologies, Inc., California, USA) to dissociate into single cells (Fig. 1a). The co-cultured cell sheets were treated with trypsin-

EDTA solution (FUJIFILM Wako Pure Chemical Corporation) and 1 mg/mL type-2 collagenase (Worthington Biomedical Corporation, Lakewood, NJ, USA) to dissociate to single cells (Fig. 2a). The cell number was counted using a hemocytometer (WakenBtech Co., Ltd, Kyoto, Japan). For the measurement of cell viability in the co-cultured cell sheet, dissociated cells were stained with Hoechst 33342 solution (FUJIFILM Wako Pure Chemical Corporation) to detect the nuclei and with Propidium Iodide (Thermo Fisher Scientific) to detect dead cells. Cells were counted using Countess 3 FL (Thermo Fisher Scientific) with Hoechst and PI staining. Cell viability (%) was calculated by dividing the number of cells detected with PI by the number of cells detected with Hoechst.

### Measurement of collagen in cell sheets

The amount of total collagen in the cell sheets was evaluated using the Collagen Quantitation Kit (Cosmo Bio Co., Ltd). Cell sheets were detached from the temperature-responsive plate on day 5 and sonicated for approximately 30 min before the measurement. The fluorescence values of the collagen calibration curves and the target sample were measured using a Nivo multimode plate reader (Perkin Elmer, Tokyo, Japan).

### Flow cytometry assessment of cell size in the cell sheet

The co-cultured cell sheets were treated with trypsin-EDTA solution (FUJIFILM Wako Pure Chemical Corporation) and 0.5 mg/ml type II collagenase (Worthington Biomedical Corporation) to prepare cell suspensions and used for flow cytometry analysis (Gallios; Beckman Coulter, Inc., Tokyo, Japan). The percentages of GFP-HUVECs in the cell sheets were measured by the intensity of green fluorescence. Forward scatter of hASCs and GFP-HUVECs were measured and analyzed by Kaluza Analysis 2.1 (Beckman Coulter, Inc.). Detailed flow cytometry strategy is provided in the Supplementary Information.

### RNA sequencing and data analysis

For the RNA sequences in Supplementary Fig. 1b, hASCs in the planar environment and hASCs in the cell sheet were analyzed. For the RNA sequence in Fig. 5, hASCs and GFP-HUVEC co-cultured cell sheets under IPP (+) and IPP (−) were analyzed (Fig. 2a). All specimens were stored at −80 °C immediately after dissolution by ISOGEN (NIPPON GENE CO., LTD., Tokyo, Japan). Subsequently, the specimens were transferred to Kyushu Pro Search LLP in Fukuoka, Japan, where RNA sequencing was

**Table 1 | Reagents used for immunohistochemistry**

| Name | Catalog No. | Dilution | Source |
|------|-------------|----------|--------|
| Anti-RFP (RABBIT) Antibody | 600-401-379 | 1:100 | Rockland Immunochemicals, Inc., Pennsylvania, USA |
| Goat anti-Rabbit IgG (H + L) Highly Cross-Adsorbed Secondary Antibody, Alexa Fluor™ 568 | A-11036 | 1:100 | Thermo Fisher Scientific Inc., Tokyo, Japan |
| CoraLite® 594-conjugated INS Monoclonal antibody | CL594-66198 | 1:200 | Proteintech Group, Inc., Tokyo, Japan |
| CD31 (PECAM-1) (89C2) Mouse mAb (Alexa Fluor® 488 Conjugate) | 42777S | 1:200 | Cell Signaling Technology, Inc. Danvers, USA |
| Hoecst 33258 | 94403 | 1:300 | Merck, Darmstadt, Germany |

*RFP* Red Fluorescent Protein, *IgG* immunoglobulin G, *CD* cluster of differentiation.

performed following the manufacturer's protocol. Total RNA was extracted from each sample using TRIZOL reagent (Thermo Fisher Scientific). RNA concentration measurements were performed using a spectrophotometer NanoDrop 2000 (Thermo Fisher Scientific). Quality confirmation was performed using the microchip electrophoresis device Bioanalyzer (Agilent Technologies Japan, Ltd., Tokyo, Japan) and the tape electrophoresis system TapeStation (Agilent Technologies Japan, Ltd.). Library preparation was performed using 100 ng of total RNA from each sample. Ribosomal RNA was removed from the total RNA using the NEBNext rRNA Depletion Kit (Human/Mouse/Rat) (E6310) (New England Biolabs Japan Inc., Tokyo, Japan) as per the reagent package insert. Libraries were prepared using the NEBNext Ultra II Directional RNA Library Prep Kit for Illumina (E7760) (New England Biolabs Japan Inc.) following the reagent package insert. The quality of the library was assessed using the Bioanalyzer DNA-sensitivity kit (5067-4626) (Agilent Technologies Japan, Ltd.). The Nextseq 500 (Illumina Inc., San Diego, USA) was used as a next-generation sequencer, and the High-output kit v2 2 × 36 PE was used as the RUN reagent. Log2 fold changes were calculated from the Generalized Linear Model (GLM), which corrects for differences in library size between the samples and effects of confounding factors. Log2 fold changes were used to compare gene expression levels. Genes with all CPM values ≤ 0.5 were considered to have negligible expression and were labeled as "N.D." (Not Detected) in the figures. For genes with CPM values of 0, a pseudo-value of CPM = 0.1 was assigned instead of 0 to enable the calculation of the standard error.

### Cultivation of co-cultured cell sheets in a mild hypoxia environment

The initial steps of harvesting co-cultured cell sheets and reattaching them to culture dishes followed standard procedures used in other experiments. Subsequently, the cell sheets were cultured under mild hypoxic conditions. Mild hypoxia cultivation was conducted using a multi-gas $CO_2$ incubator (MCO-5MUV-PJ; Panasonic Corporation, Osaka, Japan). The oxygen concentration within the incubator was maintained at 12%, and the $CO_2$ concentration at 5%. For the IPP (+) group, the cell sheets were cultured under mild hypoxic conditions with intermittent positive pressure applied, while the IPP (−) group was cultured without intermittent positive pressure in the same mild hypoxia incubator.

### Evaluation of the effect of intermittent positive pressure using NOS inhibitor

The NOS inhibitor L-NAME HCl (Selleck, Kanagawa, Japan) was used. After the co-cultured cell sheets were harvested and reattached to culture dishes, L-NAME was added to the medium in the L-NAME (+) group to achieve a final concentration of 1 mM. In the L-NAME (−) group, PBS of the same volume as L-NAME was added to the medium instead. The L-NAME (−) and L-NAME (+) groups were each cultured under either IPP (−) or IPP (+) conditions for 5 days, after which the formation of the vascular endothelial network of GFP-HUVECs and the thickness of the cell sheets were evaluated. The evaluation of the vascular endothelial network and cell sheet thickness was performed following the methods described previously.

### Measurement of dissolved oxygen during intermittent positive pressurization at the bottom of the culture dish

Air saturation at the bottom of the dish was measured using a multi-channel oxygen meter (OXY-4 mini; PreSens Precision Sensing GmbH, Regenburg, Germany) and Oxygen sensor tip (PreSens Precision Sensing GmbH) (Fig. 4a). To ascertain that the cell sheets were securely attached to the dishes, measurements were initiated one day after the cell sheets were reattached to the dishes. The medium in the dish was refreshed and placed in an incubator at 37 °C until the dish temperature stabilized at 37 °C. Once the dish temperature stabilized, pressurization was initiated for the IPP (+) group, and air saturation was measured for both the IPP (−) and IPP (+) cultured dishes.

### Transplantation of a cell sheet into athymic rats

Fourteen male athymic rats (F344/NJcl-rnu/rnu; 180–260 g; 8–11 weeks of age; CLEA Japan, Tokyo, Japan) were used for experiments shown in Fig. 7. The rats were divided into two groups: IPP (−) group (n = 7) and IPP (+) group (n = 7). After the rats were anesthetized with 2–4% inhaled isoflurane, a cell sheet cultured under IPP (−) or IPP (+) for 5 days was transplanted onto the superficial layer of the muscle (Fig. 7a). The transplantation was conducted using the cell sheet carrier material ATTRAN (WakenBtech Co., Ltd). Two days after the transplantation, the blood perfusion imaging of the transplanted cell sheet was performed and blood serum samples were collected.

### Evaluation of the transplanted tissue

The transplanted cell sheet was exposed and excised along with the superficial gluteal muscle. The cell sheet was fixed in 4% paraformaldehyde and routinely processed into 7 μm-thick paraffin-embedded sections. The sections were immunostained for Red Fluorescent Protein. The sections were incubated with primary antibodies for 2 h at room temperature and then with an appropriate secondary antibody for 40 min at room temperature (Table 1). The cross-sectional area of the immunostained cell sheet was measured using BZ-H4C analysis software (Keyence Corporation). The thickness of the cell sheets was evaluated with the same procedures as the cell sheets attached to the dishes.

### Perfusion ratio of the transplanted cell sheet

The laser Doppler perfusion system (MoorLDI2-IR; Moor Instruments, Devon, UK) was used to acquire blood perfusion imaging of the cell sheet. The perfusion ratio was calculated by dividing the perfusion value of the cell sheet region by that of the region without the cell sheet. The region outside of the cell sheet was located in the superficial layer of muscle on the same side as the transplanted cell sheet.

### Measurement of luminescence intensity of GLuc secreted by cell sheet transplanted into rat

The luminescence activity of GLuc in each 20 μL sample was measured by adding 100 μL of coelenterazine (CTZ)-containing buffer provided in the luciferase assay kit (Cosmo Bio Co., Ltd). The light intensity was measured with a Nivo multimode plate reader (Perkin Elmer).

### Insulin-GLuc secretion test for tri-cultured cells including iGLS in planar condition

Tri-cultured cells consisting of iGLs, GFP-HUVECs, and hASCs were cultured in a planar environment under IPP (−) or IPP (+) conditions for 3 days (Fig. 8a). The cells were washed with pre-warmed PBS and pre-incubated for 1 h in 2 mM glucose-KRH buffer (Cosmo Bio Co., Ltd). After the pre-incubation, the cells were incubated for 30 min in 2 mM glucose-KRH buffer followed by another 30 min in 20 mM glucose-KRH buffer Cosmo Bio Co., Ltd. After each incubation, the medium was collected and centrifuged at $\times 800 \times g$ for 5 min. The luminescence activity of insulin-GLase was measured in each 20 μL sample by adding 100 μL of CTZ. The light intensity was measured with a Nivo multimode plate reader (Perkin Elmer). The index for glucose-stimulated insulin secretion was calculated by dividing the luminescence intensity obtained from iGL stimulated with 20 mM glucose-KRH buffer by that obtained from iGL stimulated with 2 mM glucose-KRH buffer (Fig. 8c).

### Immunostaining of tri-cultured cells

Immunostaining was applied on day 3 (Fig. 8a, b). The cells were fixed with 4% paraformaldehyde and treated with 0.5% Triton for 5 min. Next, Blocking One Histo (Nacalai Tesque, Kyoto, Japan) was applied to the cells for 10 min. Finally, the cells were subjected to immunofluorescence staining (Table 1).

### Statistics and reproducibility

All statistical analyses were conducted using appropriate tests based on the characteristics of the data. For comparisons involving the same cell population under different culture conditions, data were treated as paired. For comparisons between different animals (Fig. 7j, l), data were treated as unpaired. The normality of the data was assessed using the Shapiro–Wilk test, with a significance threshold of 0.05. Equal variances were evaluated using the F-test, also with a significance threshold of 0.05. For paired datasets, normality was confirmed for all figures except for Figs. 1h, 3e, h, and 6b, c. Paired $t$-tests were applied to datasets with normal distribution. For Figs. 1h, 3e, h, and 6b, c which did not meet the normality, the Wilcoxon signed-rank test was used. For unpaired datasets, Fig. 7j met both normality and equal variances, allowing the use of the Student's $t$-test. In contrast, Fig. 7l did not meet the normality assumption, and therefore, the Mann–Whitney U test was applied. The effect size was calculated using Cohen's $d$ to evaluate the magnitude of differences between IPP (−) and IPP (+). The interpretation of Cohen's d followed the conventional thresholds: 0.2 indicates a small effect size, 0.5 indicates a moderate effect size, and values greater than 0.8 indicate a large effect size[44]. These statistical methods were chosen to ensure the appropriate analysis of the data based on their distribution and variance properties.

### Reporting summary

Further information on research design is available in the Nature Portfolio Reporting Summary linked to this article.

### Data availability

All data supporting the findings of this study are included within the article (and its supplementary files). Additional data are available from the corresponding author upon reasonable request. All RNA-seq data have been uploaded to the Sequence Read Archive (SRA) under the project accession number PRJNA1163514. Source data for each analysis can be obtained in Supplementary data 1.

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

## Acknowledgements

This research was supported by the JSPS KAKENHI Grant Numbers JP21K12654, JP22H03944, JP23K25198. We also thank Mr. Allan Nisbet for his useful comments and editorial assistance.

## Author contributions

M K, J H, Y H, H S and T S were involved in the design of the experiments. M K conducted the experiments. M K, J H and H S analyzed the data. M K, J H, H S, T S wrote the paper. M K, J H, H S, T S initiated the project. All authors discussed the results, commented on the paper and approved the final version.

## Competing interests

Tokyo Women's Medical University received research funding from CellSeed Inc. and Tokaihit Co., Ltd. Tatsuya Shimizu is a shareholder of CellSeed Inc. The other authors have no financial conflicts or competing interests to disclose.
