## [Transparent Peer Review file · Communications Biology]

Densely Vascularized Thick 3D Tissue Shows Enhanced Protein Secretion Constructed with Intermittent Positive Pressure

Corresponding Author: Dr Jun Homma

Version 0:

Reviewer comments:

Reviewer #1

(Remarks to the Author)

The manuscript provides an interesting investigation into the effects of intermittent positive pressure (IPP) on the co-culture of mesenchymal stem cells (MSCs) and vascular endothelial cells (vECs), exploring its potential for promoting vascular network formation and tissue thickening. The study presents intriguing results both in vitro and in vivo, suggesting that IPP could enhance tissue thickening and viability for transplantation purposes. While this is a relevant and important study for tissue engineering, it does not show significant advancement compared to previous works indicated in the discussion (Yoshino et al, Sekiya et al, Morrow et al). Limited investigations were conducted to explain the mechanisms that support the conclusions, and few insights were provided. More investigation is needed to develop a comprehensive understanding of why the observed effects occur, and additional experiments should be carried out to demonstrate the clinical potential. I hope the following suggestions help the authors refine their work.

Major Comments:

In Vitro:

1. Mechanisms Behind Tissue Thickening: One of the key findings is the thickening of the cell sheet, which the authors attribute to IPP-induced shear stress stimuli, in contrast to previous work by Kojima et al. However, it remains unclear whether this thickening is primarily due to mechanical stress, particularly shear stress, or, though less likely, enhanced oxygen penetration. The study would benefit from further investigation to clarify the underlying mechanism. For instance, experiments that inhibit shear stress signaling and examine its effect on collagen deposition could help distinguish between these two potential causes.

2. Novelty of Findings: While the thickening of the cell sheet is a notable result, many of the other findings (such as increased total cell number, junctions, and collagen weight) appear to be closely related to the increase in tissue volume rather than direct mechanical or metabolic effects as a result of IPP. It would be helpful to highlight more distinct, novel insights related to IPP's specific impact beyond tissue thickening.

3. Oxygen's Role: The study reports only minor differences in oxygen levels between the IPP+ and IPP- groups, making it difficult to confidently attribute changes in HIF1A and the L/G ratio to enhanced oxygen penetration. I suggest conducting additional experiments to better understand oxygen's role in tissue thickening and metabolic changes, perhaps by modulating oxygen levels independently of shear stress (e.g., using a thinner media layer, changing or mixing the media frequently, or using a higher oxygen environment).

4. Statistical Rigor: While the data on tissue thickening and related measurements are robust, several other in vitro measurements lack statistical significance, possibly due to a limited sample size or smaller cell sheets (e.g., perfusion ratio, GFP-HUVECs/total cell count %, etc.). Increasing the sample size could help clarify whether these trends of differences are meaningful or not. Similarly, it may be valuable to reassess whether certain results that the authors suggest are different (e.g., relative oxygen ratios at 48h) would remain statistically significant with more replicates.

In Vivo:

5. Mechanisms in Tricultured Condition: The higher glucose-stimulated insulin secretion index observed in the tricultured condition is interesting, but the mechanism behind this result is not fully explained. It would be helpful to clarify whether this is linked to the increased tissue thickness or other factors. Providing a more detailed analysis could also enhance the study's clinical relevance.

Minor Comments:

1. RNA-Seq Data: Including raw data and error bars for the RNA-seq analysis would improve the transparency of the results.

2. Quantification of iGLs: The increased number of iGLs needs more statistical support. Showing a single representative image does not provide a complete picture. Including data from multiple independent experiments would be more convincing.
3. Clarification of 'n': In the figure legends, please clarify what 'n' represents (e.g., fields of view, biological replicates, samples) to avoid any ambiguity.

Reviewer #2

(Remarks to the Author)

In this study, Homma et al. investigate the effects of intermittent positive pressure (IPP) on endothelial cell/ASC monolayers and vascularized cell sheet formation. The study has promise, but the following minor revisions are needed:

1. In the abstract, please specify the proteins mentioned in the sentence: "In in vivo experiments, cell sheets cultured under IPP secreted greater quantities of protein." If space allows, consider including details about islet work.
2. In the introduction, MSCs are introduced, but ASCs are used in the experiments. Please provide a description of ASCs to ensure logical flow.
3. Please provide cell counts for each cell type in the monolayer platform after 3 days of IPP- and IPP+ treatment, similar to the analysis performed for the cell sheet platform.
4. The comparison between network junctions and length in the cell sheet versus planar groups could be misleading. Since cell sheets shrink while planar cells do not, the density of the network will increase as the area decreases. Please emphasize that your analysis is based on junction and length density per mm², not total values, to avoid potential misinterpretation.
5. To support the previous point, consider showing comparison images of planar cells, and images of cell sheets at Day 1 (immediately after detachment) and Day 5 (to show shrinking). Include quantification of cell sheet sizes.
6. The manuscript's English needs thorough revision. For instance, on line 218, "more" should be removed.
7. Please double-check Figures 4c and 4d. The IPP group appears twice as thick as the control group, but the statistical significance is not as high. Verify whether the magnification is correct or if the images are representative.
8. The authors conclude that IPP increases cell viability despite finding fewer cells after dissociation compared to the initial seeding. Since cell loss is common during the dissociation of 3D tissues, the thicker tissue observed in the IPP group may not necessarily indicate better cell viability. Cells in both conditions could have proliferated, but some might have been damaged or remained unreleased from the matrix during dissociation, resulting in a lower cell count than initially seeded. If the authors wish to attribute the thicker cell sheet to improved viability, they should stain for cell death markers or analyze relevant gene expression from RNA-seq data.
9. The differences in air saturation data are minimal, particularly in the first and second experiments. The authors should improve their interpretation of these data.
10. RNA-seq data analysis could be expanded, including pathway analysis related to hypoxia and cell death/apoptosis.
11. The RNA-seq data comes from a mixed population of ASCs and ECs, so it may not be appropriate to draw firm conclusions about pathways, such as HIF1A and angiogenesis by shear stress, based on the overall RNA signal. For instance, ECs may have higher HIF1A expression, while ASCs may have lower expression, leading to an averaged result that does not reflect the individual cell types. Please discuss this limitation and consider how it might affect the interpretation of your findings.
12. Adding more details to the methods section, such as the purpose of adding ASA (promoting collagen secretion), would benefit readers.
13. GlutaMAXTM, "TM" is trademark symbol, please correct it.

Reviewer #3

(Remarks to the Author)

The authors investigated the effects of intermittent positive pressure (IPP) on co-cultured mesenchymal stem cells and vascular endothelial cells and demonstrated the potential of IPP-cultured cell sheets for transplantation therapy. In particular, they examined the protein secretion capacity and showed the usefulness of IPP for creating thicker tissue. Although these are very interesting research results, I have some points that need to be verified in detail and some modifications in the structure of the paper as shown below. I would be grateful if you could respond to them.

- (1) Please review the organization of the manuscript as a whole. The experimental conditions are consolidated in Figure 1, but it is necessary to go back and review each section individually, which hinders the understanding of the text. I think an appropriate answer is needed. Ideally, the diagram should change with each section. Excessive repetition of going back and forth from section to section is not desirable. Also, the experimental conditions (number of days of loading) are varied and the reason for these conditions is unclear.
- (2) The section "A cell sheet environment established vascular endothelial networks more effectively than a planar culture environment" in whole part mentioned to Supplementary Figures 1 and 2. These figures shouldn't be the main figure?
- (3) It seems to me that the p-values obtained from the tests are conveniently used in different ways depending on the situation. Although the standard for whether there is a significant difference or not is given, there are many cases where the explanation is given as if there is a difference even in the case of a larger p-value. This is not appropriate for scientific discussion (especially in Figure 2).
- (4) The vascular network in Figure 2a, b and Figure 3a, b are low-magnification images, so it is difficult to understand whether the network is formed or not from the images. Please provide magnified images.

- (5) Please also provide enlarged images of Figures 4c and 4e.
- (6) Regarding the results of the oxygen concentration measurement (Figure 5), it was difficult to understand what is claimed to be a reproducible result. Also, why the 48-hour endpoint data? I think you should at least provide data for each day along with the other experimental conditions.
- (7) In the section "The in vivo transplanted cocultured cell sheet cultured under intermittent positive pressure produced better protein secretion into the rat systemic circulation", it does not seem reasonable to draw such a conclusion based on the evaluation of GLuc luminescence emptiness alone.
- (8) Is the reason there are no error bars at all in the RNA-seq data because the variation is too small to see, or is there no variation at all?
- (9) It is theoretically understandable that dissolved oxygen concentration changes during IPP incubation. But what about the solubility of CO₂? Does the pH of the culture medium change? If the pH changes, this could affect the cellular response, please provide data on pH changes as well.
- (10) The RNA-seq results show changes in HIF-1alpha, is this due to changes in dissolved oxygen concentration? If so, this would be a response to changes in oxygen concentration, not a response to pressure. Would similar results be obtained in a hyperoxic environment? Or what is the mechanism of the pressure response? I think it is necessary to organize the discussion on this point.
- (11) I think it is weak to discuss the promotion of vascular endothelial network formation only by the fluctuation of RNA expression. If possible, the authors should verify the expression of representative proteins involved in the promotion of network formation among those that fluctuated.
- (12) I strongly recommend reconsideration of the assay method. I do not think it is appropriate to use the Student's t-test in a comparative test between groups of samples where equal variances cannot be assumed. Looking at the data, I do not think that equal variances can be guaranteed.

Minor points

- (1) The PDF file of the manuscript is garbled, with missing text in several places (around lines 1~20 and around lines 480~580). The author's name is missing.
- (2) The number of pages is not displayed correctly.
- (3) In the test, the effect size should be reported along with the P-value calculation.

Version 1:

Reviewer comments:

Reviewer #1

(Remarks to the Author)

My questions raised have been properly addressed by the authors through adding additional experiments and revising the manuscript.

Reviewer #2

(Remarks to the Author)

The authors have answered all my questions by performing new experiments, re-analyzing data, clarifying experimental details, and discussing the limitations. This reviewer supports the acceptance of the revised manuscript.

Reviewer #3

(Remarks to the Author)

The authors have fully addressed the points raised by the reviewers.

I would like to express my respect for the authors' efforts.

I have no additional comments.

POINT-BY-POINT RESPONSES TO THE COMMENTS OF THE REVIEWERS

Reviewer: 1

The manuscript provides an interesting investigation into the effects of intermittent positive pressure (IPP) on the co-culture of mesenchymal stem cells (MSCs) and vascular endothelial cells (vECs), exploring its potential for promoting vascular network formation and tissue thickening. The study presents intriguing results both in vitro and in vivo, suggesting that IPP could enhance tissue thickening and viability for transplantation purposes. While this is a relevant and important study for tissue engineering, it does not show significant advancement compared to previous works indicated in the discussion (Yoshino et al, Sekiya et al, Morrow et al). Limited investigations were conducted to explain the mechanisms that support the conclusions, and few insights were provided. More investigation is needed to develop a comprehensive understanding of why the observed effects occur, and additional experiments should be carried out to demonstrate the clinical potential. I hope the following suggestions help the authors refine their work.

To the reviewer:

Thank you for taking the time to review our manuscript and for providing us with valuable feedback. We have conducted several additional experiments to address your concerns and have incorporated the results into the revised manuscript. Particularly, the experiments that investigated the role of shear stress using pharmacological inhibition and those that applied Intermittent Positive Pressurization under hypoxic conditions yielded highly intriguing results. Thanks to your suggestions, we believe the quality of the manuscript has been significantly enhanced.

We hope that the revisions meet with your approval. Please note that the page and line numbers mentioned in the responses refer specifically to the main manuscript, rather than those automatically generated when the manuscript was converted to a PDF by the journal. For clarity, we have also included the revised manuscript text after each response.

Comments and responses

Major Comments:

1.Mechanisms Behind Tissue Thickening: One of the key findings is the thickening of the cell sheet, which the authors attribute to IPP-induced shear stress stimuli, in contrast to previous work by Kojima et al. However, it remains unclear whether this thickening is primarily due to mechanical stress, particularly shear stress, or, though less likely, enhanced oxygen penetration. The study would benefit from further investigation to clarify the underlying mechanism. For instance, experiments that inhibit shear stress signaling and examine its effect on collagen deposition could help distinguish between these two potential causes.

And

3.Oxygen's Role: The study reports only minor differences in oxygen levels between the IPP+ and IPP-

groups, making it difficult to confidently attribute changes in *HIF1A* and the L/G ratio to enhanced oxygen penetration. I suggest conducting additional experiments to better understand oxygen's role in tissue thickening and metabolic changes, perhaps by modulating oxygen levels independently of shear stress (e.g., using a thinner media layer, changing or mixing the media frequently, or using a higher oxygen environment).

Thank you for your important suggestions for our manuscript. Following your recommendations, we conducted additional experiments to investigate whether the thickening of the cell sheet induced by IPP is primarily due to shear stress or enhanced oxygen penetration.

To address this, we first evaluated changes in cell sheet thickness under mild hypoxia (12% O₂) in both IPP (+) and IPP (-) groups. No difference in thickness was observed between IPP (+) and IPP (-) under mild hypoxia, and neither condition showed any thickness difference compared to IPP (-) under normoxia (Figure 4b and New Figure 7a). This indicates that the enhanced oxygen penetration under normoxia is likely a key factor in cell sheet thickening.

Additionally, we examined the role of shear stress by inhibiting NOS3, a gene associated with shear stress signaling activated by IPP (Figure 6a), using the NOS inhibitor L-NAME. Even under these conditions, the IPP (+) cell sheets remained thickened (New Figure 7e). This result further supports the conclusion that shear stress alone is insufficient to induce cell sheet thickening, and that the increase in dissolved oxygen levels plays a predominant role.

Then, you raised concerns regarding about the oxygen concentration measurements (Figure 5). Our conclusions are based not only on 48-hour single time point but also on the overall trend of dissolved oxygen concentrations. To clarify our findings, we show you the graphs from three independent experiments, presented below. As demonstrated by these graphs, the dissolved oxygen concentration at the bottom of the culture medium was consistently higher in the IPP group, particularly after the first half-day of measurement. To clearly explain for readers, we have analyzed data from three additional time points—12 hours, 24 hours, and 48 hours—and incorporated these results into Figure 5c.

Additionally, to further illustrate the effect of IPP on dissolved oxygen levels in the medium, we conducted supplemental experiments measuring the oxygen content at the

bottom of the medium in the absence of a cell sheet, as shown below and added as "Supplementary figure 3". This result consistently demonstrated higher oxygen levels in the IPP group. Based on these observations—both with and without cell sheets—we are confident that IPP increases dissolved oxygen levels in the culture medium.

We believe these findings address your concerns and provide clearer insights into the mechanisms underlying the observed cell sheet thickening.

We will include these additional experimental data in the revised manuscript, as below:

Figure 5c: analyzed data from three additional time points:12 hours, 24 hours, and 48 hours

Supplementary figure 3: experiments measuring the oxygen content at the bottom of the medium in the absence of a cell sheet

Results, page 16-17, lines 5-1:

“Intermittent positive pressure does not induce thickening of the cocultured cell sheet under mild hypoxia

To investigate the effect of increased dissolved oxygen levels induced by IPP, we cultured co-cultured cell sheets under two conditions, IPP (+) and IPP (-), in a mild hypoxia environment (12% O₂, 5% CO₂), and compared their thickness. Under the mild hypoxia environment, no difference in cell sheet thickness was observed between the IPP

(+) and IPP (-) conditions [IPP (-) vs IPP (+) (μm); 33.4 ± 3.5 vs 33.5 ± 3.3] (Fig. 7a). Furthermore, there was no significant difference when compared to cell sheets cultured under normal conditions (Fig. 4b: IPP (-)), indicating that the thickening effect of IPP on cell sheets is abolished under mild hypoxia conditions.

Next, to inhibit eNOS, a phenotype of NOS3, which was upregulated by IPP (+) in the RNA-seq analysis (Fig. 6a), we added the eNOS inhibitor L-NAME and compared the thickness of co-cultured cell sheets under IPP (+) and IPP (-) conditions. The results showed that the IPP (+) group was significantly thicker than the IPP (-) group [L-NAME (+) IPP (-) vs L-NAME (+) IPP (+) (μm); 35.3 ± 4.9 vs 41.4 ± 6.3] (Fig. 7e). The inhibition of eNOS by L-NAME did not suppress the thickening effect of IPP.”

Discussion, page 24, lines 9-15:

“In the present study, under the mild hypoxia environment, IPP did not induce thickening of the co-cultured cell sheet. However, even when eNOS, one of the shear stress pathways induced by IPP, was inhibited, the cell sheet still thickened. These findings suggest that while the shear stress pathways stimulated by IPP are not limited to eNOS, shear stress alone is insufficient to induce cell sheet thickening. Instead, the increase in dissolved oxygen levels appears to play a predominant role in this process.”

Methods, page 37, lines 4–12:

“Cultivation of co-cultured cell sheets in a mild hypoxia environment

The initial steps of harvesting co-cultured cell sheets and reattaching them to culture dishes followed standard procedures used in other experiments. Subsequently, the cell sheets were cultured under mild hypoxic conditions. Mild hypoxia cultivation was conducted using a multi-gas CO₂ incubator (MCO-5MUV-PJ; Panasonic Corporation, Osaka, Japan). The oxygen concentration within the incubator was maintained at 12%, and the CO₂ concentration at 5%. For the IPP (+) group, the cell sheets were cultured under mild hypoxic conditions with intermittent positive pressure applied, while the IPP (-) group was cultured without intermittent positive pressure in the same mild hypoxia incubator.”

Methods, page 37-38, lines 14-4:

“Evaluation of the effect of intermittent positive pressure using NOS inhibitor

The NOS inhibitor L-NAME HCl (Selleck, Kanagawa, Japan) was used. After the co-cultured cell sheets were harvested and reattached to culture dishes, L-NAME was added to the medium in the L-NAME (+) group to achieve a final concentration of 1 mM. In the

L-NAME (-) group, PBS of the same volume as L-NAME was added to the medium instead. The L-NAME (-) and L-NAME (+) groups were each cultured under either IPP (-) or IPP (+) conditions for 5 days, after which the formation of the vascular endothelial network of GFP-HUVECs and the thickness of the cell sheets were evaluated. The evaluation of the vascular endothelial network and cell sheet thickness was performed following the methods described previously.”

2.Novelty of Findings: While the thickening of the cell sheet is a notable result, many of the other findings (such as increased total cell number, junctions, and collagen weight) appear to be closely related to the increase in tissue volume rather than direct mechanical or metabolic effects as a result of IPP. It would be helpful to highlight more distinct, novel insights related to IPP's specific impact beyond tissue thickening.

Thank you for providing such an important observation to enhance the quality of our manuscript. The novelty of our study lies in the “air-compressing IPP method as a technique that simultaneously enables the thickening of planar tissues and the construction of vascular networks.” As discussed in our responses to “major comments 1 and 3”, the thickening of the cell sheet is primarily attributed to the increased dissolved oxygen levels induced by air-compressing IPP.

To further investigate the role of IPP in vascular network construction, we conducted additional experiments focusing on shear stress. Using the eNOS inhibitor L-NAME, which targets NOS3, a gene associated with shear stress signaling activated by IPP (Figure 6a), we found that inhibiting eNOS did not alter the vascular network under IPP (-) conditions. However, under IPP (+), the vascular network formation was significantly suppressed to levels comparable to IPP (-) (New Figure 7b, c). These results suggest that shear stress induced by IPP contributes to vascular network construction.

Added new figure as Figure 7b, c

Moreover, our preliminary experiments demonstrated that Continuous Positive Pressure does not achieve the dual effect of cell sheet thickening and vascular network construction (detailed below). This finding underscores the uniqueness of the intermittent positive pressure (IPP) approach.

The "air-compressing IPP method as a technique that simultaneously enables the thickening of planar tissues and the construction of vascular networks" has not been previously reported with non-perfused, simplified pressurization systems. Additionally, its potential application to endocrine tissues highlights its significant novelty and developmental prospects.

We have revised the abstract to emphasize the novelty of this study, and have also updated the results and discussion sections as outlined below.

Abstract

“Constructing a dense vascular endothelial network within engineered tissue is crucial for successful engraftment. The present study investigated the effects of **air-compressing intermittent positive pressure (IPP)** on co-cultured mesenchymal stem cells and vascular endothelial cells and evaluated the potential of IPP-cultured cell sheets for transplantation therapy. The results demonstrated that the IPP (+) group exhibited a denser vascular endothelial network and significantly increased cell sheet thickness compared to the IPP (-) group. Furthermore, in vivo experiments showed that IPP-cultured cell sheets enhanced the secretion of Gaussian luciferase by genetically modified mesenchymal stem cells. **These findings highlight the IPP method as a technique that simultaneously enables the thickening of planar tissues and the construction of vascular networks.** This approach demonstrates promise for fabricating functional, transplantable, and thick tissues with dense vascularization and a high capacity for protein secretion, paving the way for novel applications in regenerative medicine.”

Results, page 17-18, lines 3-4:

“eNOS inhibition suppresses vascular network formation in co-cultured cell sheets under intermittent positive pressure

To investigate the effect of IPP on vascular endothelial network formation, we added the eNOS inhibitor L-NAME to the co-cultured cell sheets and compared vascular endothelial network formation under IPP (+) and IPP (-) conditions (Fig. 7b, c). Treatment with L-NAME under IPP (-) conditions did not lead to any changes in the formation of the vascular endothelial network compared to the group cultured under IPP(-) conditions without L-NAME [IPP (-) L-NAME (-) vs IPP (-) L-NAME (+); Total number of endothelial network junctions (count/mm²): 60.2 ± 16.6 vs 59.5 ± 14.6, Total length of the endothelial network (mm/mm²): 5.7 ± 0.8 vs 5.7 ± 0.7]. In contrast, when the cell sheets were cultured under IPP (+) conditions, the addition of L-NAME resulted in a suppression of vascular network formation compared to the group without L-NAME [IPP (+) L-NAME (-) vs IPP (+) L-NAME (+); Total number of endothelial network junctions (count/mm²): 84.1 ± 11.7 vs 67.4 ± 17.0, Total length of the endothelial network (mm/mm²): 6.7 ± 0.5 vs 6.1 ± 0.6]. Furthermore, no significant differences in vascular network formation were observed when comparing the L-NAME-treated cell sheets cultured under IPP(-) and IPP(+) conditions [IPP (-) L-NAME (+) vs IPP (+) L-NAME (+); Total number of endothelial network junctions (count/mm²): 59.5 ± 14.6 vs 67.4 ± 17.0, Total length of the endothelial network (mm/mm²): 5.7 ± 0.7 vs 6.1 ± 0.6] (Fig. 7b, c). Inhibition of eNOS by L-NAME suppressed the vascular network formation induced by IPP (+), indicating the involvement of eNOS in the process.”

Discussion, page 22, lines 5-15:

“In the present experiment, the expression of VE-cadherin increased, as did the expression of multiple related genes downstream of VE-cadherin signals (VEGFR2 gene KDR, eNOS gene NOS3, and cyclin A gene CCNA1) caused by shear stress¹⁵. Furthermore, eNOS inhibition experiments in the present study demonstrated that inhibiting eNOS under IPP (+) conditions significantly suppressed vascular network formation, bringing it to levels comparable to those observed under IPP (-) conditions. This result is consistent with the RNA-seq findings and suggests a strong correlation between vascular network formation induced by the IPP method and shear stress. In contrast, the increased expression of bFGF³⁰, TRPV4³¹, integrin α 5³², AQP1, α 1-AR, and SR-2A¹³, which have all been reported to be associated with hydrostatic stimulation, was not confirmed in the present study.”

Figure legends, page 53-54, lines 9-14:

“Figure 7: Effects of mild hypoxia on intermittent positive pressure-induced responses in co-cultured cell sheets and inhibition of shear stress pathways by L-NAME

(a) The average of thickness of cell sheets under mild hypoxia on day 5 measured using OCT images [IPP (-) vs IPP (+) (μm): $n = 6$, $p = 0.94$, Cohen's $d = 0.02$]. (b) Total number of endothelial network junctions on day 5 [IPP (-) L-NAME (-) vs IPP (-) L-NAME (+) (count/ mm^2): $n = 6$, $p = 0.56$, Cohen's $d = 0.05$], [IPP (-) L-NAME (+) vs IPP (+) L-NAME (+) (count/ mm^2): $n = 6$, $p = 0.44$, Cohen's $d = 0.50$], [IPP (+) L-NAME (-) vs IPP (+) L-NAME (+) (count/ mm^2): $n = 6$, * $p < 0.05$, Cohen's $d = 1.14$]. (c) The total length of the endothelial network on day 5 [IPP (-) L-NAME (-) vs IPP (-) L-NAME (+) (mm/mm^2): $n = 6$, $p = 0.84$, Cohen's $d = 0.06$], [IPP (-) L-NAME (+) vs IPP (+) L-NAME (+) (mm/mm^2): $n = 6$, $p = 0.31$, Cohen's $d = 0.56$], [IPP (+) L-NAME (-) vs IPP (+) L-NAME (+) (mm/mm^2): $n = 6$, * $p < 0.05$, Cohen's $d = 1.31$]. (d) Representative image of GFP-HUVECs (green) of each group on day 5. Scale bar: $500\mu\text{m}$. (e) The average of thickness of cell sheets on day 5 measured using OCT images of each group [IPP (-) L-NAME (-) vs IPP (-) L-NAME (+) (μm); 35.5 ± 4.9 vs 35.3 ± 4.9 , $n = 6$, $p = 0.91$, Cohen's $d = 0.04$], [IPP (-) L-NAME (+) vs IPP (+) L-NAME (+) (μm); 35.3 ± 4.9 vs 41.4 ± 6.3 , $n = 6$, * $p < 0.05$, Cohen's $d = 1.09$], [IPP (+) L-NAME (-) vs IPP (+) L-NAME (+) (μm); 40.4 ± 5.4 vs 41.4 ± 6.3 , $n = 6$, $p = 0.58$, Cohen's $d = 0.17$]. (a, e) $N = 6$, representing 6 cell sheets, with the average thickness measured at 3 random locations on each sheet. (b, c) $N = 6$, representing 6 cell sheets, with the evaluation of the endothelial vascular network performed in 3 randomly selected fields of view per sheet.”

4. Statistical Rigor: While the data on tissue thickening and related measurements are robust, several other in vitro measurements lack statistical significance, possibly due to a limited sample size or smaller cell sheets (e.g., perfusion ratio, GFP-HUVECs/total cell count %, etc.). Increasing the sample size could help clarify whether these trends of differences are meaningful or not. Similarly, it may be valuable to reassess whether certain results that the authors suggest are different (e.g., relative oxygen ratios at 48h) would remain statistically significant with more replicates.

Thank you for bringing this important point to our attention. Indeed, for the in vitro data in Figures 2d, 2e and 4h, it would be better to revise the interpretation of the statistical results and the description from the perspective of scientific discussion. On the other hand, as you know, in animal experiments, it is controversial to discuss results based solely on p-values. Regarding Figure 8j, while the p-value is 0.13, the large effect size (Cohen's $d = 0.96$) suggests that we cannot definitively conclude there is no difference between the IPP (+) and IPP (-) groups. But, However, we believe that the scientific benefit of additional animal experiments is limited, and we have decided not to conduct further animal experiments at this time. That

being said, we recognize that the expression as it stands could lead to statistical misunderstandings. Therefore, we have carefully revised the description, including the interpretation of the effect size, using the phrase "tend to" to avoid potential confusion and to ensure a more precise presentation.

These revisions did not affect the major findings of our paper.

Regarding **Figure 2d and 2e**

Results, page 7, lines 3-4:

Before revision (original manuscript):

“Vascular endothelial cells in planar co-cultured conditions under intermittent positive pressure tend to better form vascular networks than in an unpressurized state.”

After revision (revised manuscript):

“Impact of intermittent positive pressure on endothelial cells forming vascular networks in planar co-culture conditions.”

Results, page 7, lines 7-13:

“Over a period of three days, GFP-HUVECs infiltrated the intracellular spaces and began to connect with each other, forming a vascular network **in both the IPP (-) and (+) groups (Fig. 2b, c). The number of network junctions did not differ significantly between the IPP (-) and IPP (+) groups** [IPP (-) vs IPP (+) (count/mm²); 7.1 ± 4.1 vs 9.4 ± 2.7 , $p = 0.10$, **Cohen’s $d = 0.66$**] (Fig 2d), and the total network length in the IPP (+) **was also not significantly different** [IPP (-) vs IPP (+) (mm/mm²); 3.5 ± 1.2 vs 4.2 ± 0.7 , $p = 0.09$, **Cohen’s $d = 0.90$**] (Fig. 2e).”

Discussion, page 20, lines 12-14:

Before revision (original manuscript):

“i) IPP, which was generated by a air compressing pressurization device, stimulated promotion of vEC network formation in a coculture condition of MSCs and vECs. Moreover, the network effect was even more pronounced in cell sheets”

After revision (revised manuscript):

“i) IPP, which was generated by an air-compressing pressurization device, **promoted vEC network formation in the coculture cell sheet of MSCs and vECs.**”

Discussion, page 21, lines 10-13:

“The IPP method in the co-culture environment of the present study also **demonstrated a**

similar effect in promoting vascular network formation within a cell sheet structure, despite clear differences in the pressure application methods.”

Regarding **Figure 4h**

Results, page 12, lines 8-10:

Before revision (original manuscript):

“The proportion of GFP-HUVECs in all cells was slightly higher in the IPP (+) group ($6.2 \pm 1.0\%$) compared to the IPP (-) group ($4.3 \pm 1.5\%$) (Fig. 4h)”

After revision (revised manuscript):

“The proportion of GFP-HUVECs in all cells **was no significantly differences between** the IPP (+) group ($6.2 \pm 1.0\%$) and the IPP (-) group ($4.3 \pm 1.5\%$) (Fig. 4h)”

Regarding **Figure 8**

Results, page 18, lines 6-7:

Before revision (original manuscript):

“The in vivo transplanted cocultured cell sheet cultured under intermittent positive pressure produced better protein secretion into the rat systemic circulation.”

After revision (revised manuscript):

“Evaluation of the impact of intermittent positive pressure on the thickness and functionality of co-cultured cell sheets transplanted in Vivo.”

Results, page 18-19, lines 16-2:

“The intensity of GLuc luminescence in rat blood **tend to be** higher in the IPP (+) group [IPP (-) vs IPP (+) (AU); 2600 ± 224 vs 2824 ± 241 , $p = 0.13$, Cohen’s $d = 0.96$]: **Although the difference did not reach statistical significance ($p = 0.13$), the large effect size in the animal experience (Cohen’s $d = 0.96$) suggests a notable trend (Fig. 8j).**”

5. Mechanisms in Tricultured Condition: The higher glucose-stimulated insulin secretion index observed in the tricultured condition is interesting, but the mechanism behind this result is not fully explained. It would be helpful to clarify whether this is linked to the increased tissue thickness or other factors. Providing a more detailed analysis could also enhance the study’s clinical relevance.

We appreciate your interest in the results regarding the glucose-stimulated insulin secretion index under the triculture condition. As mentioned in the Discussion section (refer to our response to minor comment 2), we believe that the improvement in dissolved oxygen levels by IPP is involved in the observed increase in the insulin secretion index. However, this has not

been thoroughly investigated in the present manuscript. The relationship between IPP and increased insulin secretion involves several aspects requiring further investigation, such as the behavior of pancreatic β -cells and changes in cell numbers. We plan to address these issues in future studies. Nevertheless, we believe that the results of this experiment provide valuable insights into the applicability of IPP in endocrine cells, even without a definitive explanation of the mechanisms, and thus merit inclusion in this manuscript.

Minor Comments:

1. RNA-Seq Data: Including raw data and error bars for the RNA-seq analysis would improve the transparency of the results.

Thank you for your valuable comment. In response, we have revised the RNA-seq graph to make it more comprehensive and include error bars by using Log2 Fold Change (Log2FC) instead of Fold Change (FC). The error bars were calculated based on the standard error of the Log2FC for each experiment. Additionally, to enhance data transparency, we have included a table in the manuscript that provides the raw CPM data for each sample (Supplementary table 1). Furthermore, all RNA-seq data have been uploaded to the Sequence Read Archive (SRA) under the project accession number PRJNA1163514. These revisions are detailed in the updated manuscript as outlined below:

RNA-seq figure revision: Figure 6 and Supplementary figure 1
Additional Table for the raw CPM data: Supplementary Table 1

Method, page 36-37, lines 14-2:

“Log2 fold changes were calculated from Generalized Linear Model (GLM), which corrects for differences in library size between the samples and effects of confounding factors. Log2 fold changes were used to compare gene expression levels. Genes with all CPM values ≤ 0.5 were considered to have negligible expression and were labeled as “N.D.” (Not Detected) in the figures. For genes with CPM values of 0, a pseudo-value of CPM = 0.1 was assigned instead of 0 to enable the calculation of the standard error.”

Data availability statement Section, page 43, lines 9-11:

“All RNA-seq data have been uploaded to the Sequence Read Archive (SRA) under the project accession number PRJNA1163514.”

2. Quantification of iGLs: The increased number of iGLs needs more statistical support. Showing a single representative image does not provide a complete picture. Including data from multiple independent experiments would be more convincing.

Thank you for your insights regarding the iGLs-coculture experiments; indeed, there were expressions in the description of the results that could cause misunderstanding. In this experiment, we did not examine changes in the number of iGL cells. The image was intended to illustrate the observation that iGL cells appeared to cluster around endothelial cells. However, as it is difficult to draw conclusions about differences in behavior based solely on these images, we have decided to remove this description and instead present the images as those of Triculture cells in both groups. Moving forward, we plan to further investigate the relationship IPP and iGL cell behavior. Additionally, the Discussion section included descriptions related to cell death, which may have led to misunderstanding regarding changes in iGL numbers. We have revised this part along with the references. The changes are as follows:

Deleted from original manuscript:

~~“Then, under immunofluorescent observation, a greater number of iGLs were observed to be clustered around GFP HUVECs in the IPP (+) groups than in the IPP (-) group (Fig. 8e).”~~

Results, page 19-20, lines 17-2:

“With a view to the therapeutic application of culturing under intermittent positive pressure, iGL, (a clonal INS-1E cell line derived from rat pancreatic β -cells and stably expressing a fusion protein of insulin and Gaussia luciferase (insulin-GLuc)), hASCs and GFP HUVECs were tri-cultured (Fig. 9b).”

Discussion, page 26-27, lines 17-2:

Before revise

“It is well established that pancreatic islet β cells are susceptible to cell death in a hypoxic environment, which has a detrimental effect on glucose-responsive insulin secretion³⁸. Inducing aerobic metabolism in pancreatic islet β cell co-culture using IPP is an effective approach.”

After revise

“It is well established that pancreatic islet β cells are sensitive to hypoxic environments, where improved oxygenation can enhance glucose-responsive insulin secretion⁴². Improving oxygenation in pancreatic islet β cell co-culture using IPP is an effective approach.”

References, page 47, lines 14-16:

Deleted References

~~“38. Cantley, J., Grey, S. T., Maxwell, P. H. & Withers, D. J. The hypoxia response~~

pathway and β -cell function. *Diabetes Obes Metab* 12, 159–167 (2010)”

Added References

“42. Ohta, M., Nelson, D., Nelson, J., Meglasson, M. D. & Erecińska, M. Oxygen and temperature dependence of stimulated insulin secretion in isolated rat islets of Langerhans. *J Biol Chem* 265, 17525–32 (1990).”

3. Clarification of 'n': In the figure legends, please clarify what 'n' represents (e.g., fields of view, biological replicates, samples) to avoid any ambiguity.

Thank you for your recommendation. As you suggested, we have added a description to the all figure legends to clarify what 'n' represents.

For example:

“Figure 2. (d) The total number of endothelial network junctions on day 3 ($n = 8$, measured from 8 independent culture dishes, with the mean of 3 randomly selected fields of view per dish, $p = 0.10$, Cohen’s $d = 0.66$).”

Reviewer #2 (Remarks to the Author):

In this study, Homma et al. investigate the effects of intermittent positive pressure (IPP) on endothelial cell/ASC monolayers and vascularized cell sheet formation. The study has promise, but the following minor revisions are needed:

To the reviewer:

Thank you for taking the time to review our manuscript and for providing us with positive feedback. We have performed some additional experiments to address your concerns, and we have added the results to the revised manuscript. We hope that the revisions we have made meet with your approval. Please note that the page and line numbers stated in the responses are those specifically generated for the main manuscript rather than those automatically generated when the manuscript was converted to a PDF by the journal. For clarity, we have also provided the revised manuscript text after each response.

Comments and responses

1. In the abstract, please specify the proteins mentioned in the sentence: "In in vivo experiments, cell sheets cultured under IPP secreted greater quantities of protein." If space allows, consider including details about islet work.

Thank you for your recommendation. As suggested, we have specified the proteins (in red) in the abstract. However, including the "islet work" within the 150-word limit proved challenging, so we chose to omit it. Additionally, the abstract has been revised to reflect the other changes made in this revision.

Abstract

After revision

"Constructing a dense vascular endothelial network within engineered tissue is crucial for successful engraftment. The present study investigated the effects of air-compressing intermittent positive pressure (IPP) on co-cultured mesenchymal stem cells and vascular endothelial cells and evaluated the potential of IPP-cultured cell sheets for transplantation therapy. The results demonstrated that the IPP (+) group exhibited a denser vascular endothelial network and significantly increased cell sheet thickness compared to the IPP (-) group. Furthermore, **in vivo experiments showed that IPP-cultured cell sheets enhanced the secretion of Gaussian luciferase by genetically modified mesenchymal stem cells.** These findings highlight the IPP method as a technique that simultaneously enables the thickening of planar tissues and the construction of vascular networks. This approach

demonstrates promise for fabricating functional, transplantable, and thick tissues with dense vascularization and a high capacity for protein secretion, paving the way for novel applications in regenerative medicine.”

2. *In the introduction, MSCs are introduced, but ASCs are used in the experiments. Please provide a description of ASCs to ensure logical flow.*

Thank you for your constructive advice. As you mentioned we added a description of ASCs and a reference to ensure logical flow as below:

Introduction, page 4, lines 7-12:

“Several studies have demonstrated that MSCs secrete various ECMs, which have been linked to tissue regeneration, self-renewal, and stem cell differentiation, making it useful for tissue engineering¹⁰. Among MSCs, adipose tissue-derived MSCs (ASCs) have been reported to exhibit superior angiogenic effects¹¹. Therefore, with the goal of promoting a vascular endothelial network in vitro, co-culturing MSCs, particularly ASCs, with vECs is presumed to be an effective approach to promote pro-angiogenic factors and ECMs.”

Reference, No11

“11. Hsiao, S. T.-F. et al. Comparative Analysis of Paracrine Factor Expression in Human Adult Mesenchymal Stem Cells Derived from Bone Marrow, Adipose, and Dermal Tissue. *Stem Cells Dev* 21, 2189–2203 (2012).”

3. *Please provide cell counts for each cell type in the monolayer platform after 3 days of IPP- and IPP+ treatment, similar to the analysis performed for the cell sheet platform.*

As per your recommendation, we have added cell count analysis data for each cell type in the monolayer platform after 3 days of IPP- and IPP+ treatment, which is now included in the revised Figure 2h. The results show that the HUVEC ratio exhibits no significant differences between IPP- and IPP+ treatments, consistent with the findings in the cell sheet platform. In relation to this addition, we have revised the “*Results*” section and the “*Figure Legends*” as follows:

Results, page 8, lines 10-12:

“The proportion of GFP-HUVECs among all cells showed no significant difference between the IPP (-) group ($7.4 \pm 3.4\%$) and the IPP (+) group ($7.1 \pm 3.7\%$) (Fig. 2h).”

Figure legends, page 49-50, lines 17-1:

“(h) The percentage of GFP-HUVECs of the total cell number in the co-cultured cells

in planar conditions (n = 9, independent samples, $p = 0.50$, Cohen's $d = 0.10$).”

4. The comparison between network junctions and length in the cell sheet versus planar groups could be misleading. Since cell sheets shrink while planar cells do not, the density of the network will increase as the area decreases. Please emphasize that your analysis is based on junction and length density per mm^2 , not total values, to avoid potential misinterpretation.

As per your recommendation, we have emphasized that the evaluation is based on density per mm^2 to avoid potential misinterpretation due to differences in cell density. Additionally, to highlight the characteristics of the vascular endothelial cell network in the cell sheet compared to planar cultures, we have included co-cultured images of both the day 5 planar culture and the cell sheet in Supplementary Fig. 2c. These figures show that, by day 5, the vascular network in the planar culture had deteriorated, whereas the vascular network in the cell sheet was maintained. Based on the network analysis and these images from day 5, we revised the statement in the manuscript from “A promotion of the vascular endothelial network of the co-cultured vECs was more pronounced in the cell sheet environment” to “The cell sheet environment enables the construction and maintenance of a denser vascular network.” The revised text is as follows:

Results, page 9-10, lines 9-4:

“A promotion of the vascular endothelial network of the co-cultured vECs was more pronounced in the cell sheet environment than in the planar environment (Supplementary Fig. 2c). On day 3, The vascular endothelial network was compared between planar cultures and cell sheets (Supplementary Fig. 2d and 2e). As co-cultured cell sheets shrink immediately after detachment from the temperature-responsive culture dish, the analysis was conducted using the unit area “per mm^2 ” to account for differences in cell density between the two groups. Both the number of network junctions and the length of the entire network were significantly greater in cell sheet environment than the planar culture conditions [cell sheet vs planar culture: (count/ mm^2); 12.4 ± 3.2 vs 128.1 ± 29.3 ”

(Supplementary Fig. 2d), (mm/mm^2); 3.9 ± 0.5 vs 17.2 ± 1.8 (Supplementary Fig. 2e)]. Furthermore, by day 5, the vascular network in planar cultures had deteriorated, whereas the vascular network in the cell sheet environment was well-maintained (Supplementary Fig. 2c). These results demonstrated that the cell sheet environment facilitated the formation of a dense vascular endothelial network and contributed to its sustained maintenance over 5 days.”

We have included co-cultured images of both the day 5 planar culture and the cell sheet; please refer to Supplementary Fig. 2c and its figure legend.

5. To support the previous point, consider showing comparison images of planar cells, and images of cell sheets at Day 1 (immediately after detachment) and Day 5 (to show shrinking). Include quantification of cell sheet sizes.

Thank you for your insights regarding the morphological changes of the cell sheet; however, there may have been some differences in interpretation. The cell sheet floats up while shrinking during detachment from the temperature-responsive culture dish. Therefore, the shrinkage of the cell sheet we refer to occurs immediately after detachment, not as a gradual shrinking over the culture period. In the original manuscript, the description regarding cell sheet harvesting may have been unclear, so we have revised it. Additionally, to aid in the understanding of the cell sheet, we have included a reference to our paper showing the thickening of the cell sheet directly after detachment. We have also added images of the cell sheet and a comparison of the sheet area immediately after detachment in Supplementary figure 2a and 2b.

Results, page 8-9, lines 16–3:

“A cell sheet is a tissue with a higher density than a planar culture dish²⁰. When cells are seeded on a temperature-responsive culture dish and harvested as a cell sheet, the cells that were adhered to the dish begin to shrink as they detach. Immediately after detachment (Supplementary Fig. 2a), the cell sheet exhibits an area that is 12 times smaller than in the planar culture state (Supplementary Fig. 2b). As a result, the cell sheet tissue becomes

thicker and easier to handle as a planar cell tissue.”

Method, page 31, lines 16-18:

“Imaging devices

An image of a co-cultured cell sheet presented in Supplementary Figure 2a was captured using an iPhone 12 mini (Apple Inc., Cupertino, CA, USA).”

Added References

“20. Homma, J., Shimizu, S., Sekine, H., Matsuura, K. & Shimizu, T. A novel method to align cells in a cardiac tissue-like construct fabricated by cell sheet-based tissue engineering. *J Tissue Eng Regen Med* **14**, 944–954 (2020).”

6. *The manuscript's English needs thorough revision. For instance, on line 218, "more" should be removed.*

Thank you for your suggestion. As part of this revision, we thoroughly reviewed the manuscript and made the necessary corrections throughout. We appreciate your attention to detail.

7. *Please double-check Figures 4c and 4d. The IPP group appears twice as thick as the control group, but the statistical significance is not as high. Verify whether the magnification is correct or if the images are representative.*

Thank you for your constructive feedback. As you pointed out, the original images were not the most representative of our statistical results. Specifically, we had included examples that highlighted the most distinct differences between the two conditions. In response, we have replaced them with section images that better align with the overall data and provide a more accurate representation. Additionally, we have included magnified images to enhance clarity and ensure the data is presented as transparently as possible: Figure 4c and 4d.

8. *The authors conclude that IPP increases cell viability despite finding fewer cells after dissociation compared to the initial seeding. Since cell loss is common during the dissociation of 3D tissues, the thicker tissue observed in the IPP group may not necessarily indicate better cell viability. Cells in both conditions could have proliferated, but some might have been damaged or remained unreleased from the matrix during dissociation, resulting in a lower cell count than initially seeded. If the authors wish to attribute the thicker cell sheet to improved viability, they should stain for cell death markers or analyze relevant gene expression from RNA-seq data.*

Thank you for your constructive query. In this revision, we conducted additional experiments

to measure cell viability using the cell death marker propidium iodide (PI). As shown in the new figure below (New Figure 4i), cell viability was significantly higher in the IPP group. This result reinforces the cell count findings (Figure 4g) and provides additional support for our conclusions.

Results, page 12, lines 10-11:

“The cell viability of the total cells on day 5 was significantly higher in the IPP (+) group (85.0 ± 7.1%) compared to the IPP (-) group (69.1 ± 7.3%) (Fig 4i).”

Methods, page 34, lines 9-14:

“For the measurement of cell viability in the co-cultured cell sheet, dissociated cells were stained with Hoechst 33342 solution (FUJIFILM Wako Pure Chemical Corporation) to detect the nuclei and with Propidium Iodide (Thermo Fisher Scientific) to detect dead cells. Cells were counted using Countess 3 FL (Thermo Fisher Scientific) with Hoechst and PI staining. Cell viability (%) was calculated by dividing the number of cells detected with PI by the number of cells detected with Hoechst.”

Figure legend, page 52, lines 2-4:

“(i) The percentage of cell viability in the co-cultured cell sheets on day 5 (n = 5, independent samples, ** $p < 0.01$, Cohen’s $d = 2.2$).”

9. The differences in air saturation data are minimal, particularly in the first and second experiments. The authors should improve their interpretation of these data.

Thank you for your important suggestion regarding the results of the oxygen concentration measurements (Figure 5). As you pointed out, it may indeed be difficult to discern the differences in dissolved oxygen levels between the IPP and control groups based solely on the data at the 48-hour time point. However, our conclusions are based not only on this single time point but also on the overall trend of dissolved oxygen concentrations. To clarify our findings, we show you the graphs from three independent experiments, presented

below. As demonstrated by these graphs, the dissolved oxygen concentration at the bottom of the culture medium was consistently higher in the IPP group, particularly after the first half-day of measurement. However, we understand your concern about relying solely on the 48-hour time point to present the differences in oxygen content. Therefore, we have shown data from three additional time points—12 hours, 24 hours, and 48 hours—and incorporated these results into Figure 5c for better clarity.

Additionally, to further illustrate the effect of IPP on dissolved oxygen levels in the medium, we conducted a supplemental experiment measuring the oxygen content at the bottom of the medium in the absence of a cell sheet, as shown below and added as "Supplementary figure 3". This result consistently demonstrated higher oxygen levels in the IPP group. Based on these observations—both with and without cell sheets—we are confident that IPP increases dissolved oxygen levels in the culture medium.

We included these additional experimental data in the revised manuscript, as below:

Figure 5 C: analyzed data from three additional time points: 12 hours, 24 hours, and 48 hours

Supplementary figure 3: experiments measuring the oxygen content at the bottom of the medium in the absence of a cell sheet

10. RNA-seq data analysis could be expanded, including pathway analysis related to hypoxia and cell death/apoptosis.

Thank you for your inquiry regarding the RNA-seq analysis. In this revision, we reanalyzed the RNA-seq data and modified the description (changing Fold Change to Log2 Fold Change) and classifications in Figure 6. Specifically, we added graphs focusing on cell cycle-related genes (New Fig. 6b) and hypoxia-inducible related genes (New Fig. 6d). Below is a summary of the findings:

Cell cycle-related genes (New Fig. 6b): There were no significant differences in the expression of apoptosis-related genes (CASP3, CASP8, CASP9, BCL1, BCL2, and BAX). However, analysis of genes associated with cell proliferation and cell cycle progression revealed upregulation of NOTCH1, NOTCH3, NOTCH4 signaling and their downstream targets, including MYC, CCND1, CDK6, and CCNA1, in the IPP (+) group. These genes, particularly MYC, are known to be related to cell cycle progression of MSCs²⁵.

Furthermore, the NOTCH signaling pathway plays a critical role as a mechanosensor for shear stress in endothelial cells²³, suggesting its involvement in the increased cell numbers observed in the IPP (+) group compared to the IPP (-) group, as well as in the formation of vascular endothelial networks. This pathway likely represents a key mechanism underlying the findings of our study, and we plan to further investigate it in future research.

Hypoxia-inducible related genes (New Fig. 6d): Among the HIF family members, HIF1 α (gene: HIF1A), which is known to increase under hypoxia, showed significantly decreased expression in the IPP (+) group. Similarly, ARNT and ARNT2 (HIF-1 β), genes associated with HIF1A, were also significantly downregulated under IPP (+) conditions. To provide a clearer overview of hypoxia-related gene expression, we created a new graph (Fig. 6d) in Figure 6, which includes RNA-seq data for additional genes, such as HIF2 α , HIF3 α , and HIF1 β .

Based on these additional RNA-seq analyses, we have revised the Results, Discussion and References sections, as below:

Results

page 13-14, lines 12–15

“Comparison RNA expression related to the promotion of vascular endothelial networking in co-cultured cell sheets under intermittent positive pressure and non-pressurization.

In order to gain insight into the differences in vascular endothelial network promotion between the IPP (-) and IPP (+) from the perspective of gene expression, total RNA sequencing of the co-cultured cell sheet was performed (Fig. 3a). In the present study, the results were confirmed in tests of mechanotransduction, **cell cycle**, ECM, **Hypoxia-inducible factor (HIF) family** and angiogenesis paracrine factor.

Figure 6a shows the expression of certain RNA related to mechanotransduction^{21,22}. The

genes encoding Notch receptor 1 (NOTCH1), Notch receptor 3 (NOTCH3) and Notch receptor 4 (NOTCH4), which are expressed in vECs and associated with shear stress²³, exhibited increases of 0.6-, 0.5-, and 0.3-Log₂ fold changes, respectively. Additionally, P2X4 receptor (P2RX4), an ion channel that senses stimulation through mechanotransduction caused by shear stress²⁴, demonstrated an increase of 0.4-Log₂ fold change in the IPP (+) group. Furthermore, genes related to downstream signaling of mechanotransduction under shear stress, including the VE-cadherin gene (CDH5), the VEGFR2 gene (KDR) and eNOS gene (NOS3) showed increased expression in the IPP (+) group (0.9-, 0.6-, 1.0- and 1.0-Log₂ fold changes, respectively) (Fig. 6a). These results suggest that IPP caused mechanotransduction stimuli to the cells.

Figure 6b shows the expression of certain RNA associated with cell cycle regulation. The RNA expression of the MYC gene (MYC), cyclin D1 gene (CCND1), cyclin-independent kinase 6 gene (CDK6), and cyclin A1 gene (CCNA1), all of which are related to cell cycle progression and cell proliferation in mesenchymal stem cells²⁵, were increased in the IPP (+) group (0.5-, 0.3-, 0.3- and 1.4-Log₂ fold changes, respectively). As these genes are downstream targets of Notch receptor signaling, the results suggest a potential link between IPP and enhanced cell proliferation.”

page 15, lines 7-13:

“Next, Figure 6d shows the expression of RNA associated with the HIF family. Among the genes related to the HIF family, the HIF-1 α gene (HIF1A) and the HIF-1 β gene (ARNT) exhibited decreased expression in the IPP (+) group, with -0.5- and -0.2-Log₂ fold changes, respectively. No significant differences were observed in the expression levels of genes related to HIF-2 α (EPAS1) and HIF-3 α (HIF3A). These results suggest a potential relationship between IPP (+) and the inhibition of the HIF pathway.”

Discussion, page 25, lines 1-11:

“Additionally, in the present study, the number of cells under IPP culture was found to be greater than under non-IPP culture. RNA-seq analysis revealed the upregulation of NOTCH1, NOTCH3, NOTCH4 signaling and their downstream targets, including MYC, CCND1, CDK6, and CCNA1, in the IPP (+) group. These genes, particularly MYC, are known to be associated with the cell cycle progression of MSCs²⁵. Furthermore, NOTCH1 is recognized as a key component of the shear stress signaling pathway. These findings are consistent with the hypothesis that the activation of shear stress signals induced by IPP promotes cell proliferation. While the thickening of the cell sheet under IPP conditions is primarily attributed to ECM components such as collagen, the difference in cell numbers also appears to contribute to this outcome. Further investigations are required to clarify the effects of IPP-induced NOTCH signaling activation on cells.”

Reference

- “23. Mack, J. J. *et al.* NOTCH1 is a mechanosensor in adult arteries. *Nat Commun* **8**, 1620 (2017).
25. Satoh, Y. *et al.* Roles for c-Myc in Self-renewal of Hematopoietic Stem Cells. *Journal of Biological Chemistry* **279**, 24986–24993 (2004).”

11. The RNA-seq data comes from a mixed population of ASCs and ECs, so it may not be appropriate to draw firm conclusions about pathways, such as HIF1A and angiogenesis by shear stress, based on the overall RNA signal. For instance, ECs may have higher HIF1A expression, while ASCs may have lower expression, leading to an averaged result that does not reflect the individual cell types. Please discuss this limitation and consider how it might affect the interpretation of your findings.

Thank you for your insightful comment. As you pointed out, our study does not include separate analyses of ASCs and ECs within the IPP system. Therefore, it is indeed challenging to accurately represent the individual effects of ASCs and ECs in this context. However, analyzing ASCs and ECs independently would not capture the interactions between these two cell types under co-culture conditions, which are integral to understanding the effects of the IPP system. The aim of this study is not to evaluate the effects of IPP on vascular endothelial cells or mesenchymal stem cells in isolation. Instead, we sought to investigate the overall effects of IPP on the co-cultured cell sheet with vECs networks as a whole. For this reason, we focused on the RNA-seq analysis of the network sheet as a combined system of ASCs and vECs, as this approach allows us to consider their interactions and better aligns with the study's objective. That said, we agree with your observation that exploring the gene expression behavior of each cell type in the co-culture environment will be an important focus for future research. We would plan to address this by employing techniques such as single-cell RNA-seq in subsequent studies.

12. Adding more details to the methods section, such as the purpose of adding ASA (promoting collagen secretion), would benefit readers.

Thank you for your constructive advice. As you suggested, we added a description of ASA in the Methods section along with a relevant reference, as detailed below:

Method, page 30, lines 6-9:

“The cells were cultured in basal medium with 40 µg/mL of L-ascorbic acid phosphate magnesium salt n-hydrate (ASA) (FUJIFILM Wako Pure Chemical Corporation). **ASA was included because of its ability to promote collagen secretion from MSCs, which was expected to enhance the formation of well-structured cell sheets⁴³.**”

Reference, No. 43:

“43. Yu, J., Tu, Y.-K., Tang, Y.-B. & Cheng, N.-C. Stemness and transdifferentiation of adipose-derived stem cells using l-ascorbic acid 2-phosphate-induced cell sheet formation. *Biomaterials* 35, 3516–3526 (2014).”

13. GlutaMAXTM, “TM” is trademark symbol, please correct it.

Thank you for bringing these inadvertent omissions to our attention. We believe these issues arose during the automatic PDF conversion process at the time of submission. We have carefully reviewed the file generated after this conversion and submitted it accordingly.

Method, page 29, lines 10-12:

“GlutaMAXTM supplement medium (Thermo Fisher Scientific, Waltham MA, USA) containing 10% fetal bovine serum (FBS) and penicillin-streptomycin solution (×100) (FUJIFILM Wako Pure Chemical Corporation, Osaka, Japan).”

Reviewer#3(Remarks to the Author):

The authors investigated the effects of intermittent positive pressure (IPP) on co-cultured mesenchymal stem cells and vascular endothelial cells and demonstrated the potential of IPP-cultured cell sheets for transplantation therapy. In particular, they examined the protein secretion capacity and showed the usefulness of IPP for creating thicker tissue. Although these are very interesting research results, I have some points that need to be verified in detail and some modifications in the structure of the paper as shown below. I would be grateful if you could respond to them.

To the reviewer:

Thank you for taking the time to review our manuscript and for providing us with valuable feedback. We have conducted several additional experiments and analysis to address your concerns and have incorporated the results into the revised manuscript. Particularly, the experiments that investigated the role of shear stress using pharmacological inhibition and those that applied Intermittent Positive Pressurization under hypoxic conditions yielded highly intriguing results. Thanks to your suggestions, we believe the quality of the manuscript has been significantly enhanced.

We hope that the revisions meet with your approval. Please note that the page and line numbers mentioned in the responses refer specifically to the main manuscript, rather than those automatically generated when the manuscript was converted to a PDF by the journal. For clarity, we have also included the revised manuscript text after each response.

Comments and responses

(1) Please review the organization of the manuscript as a whole. The experimental conditions are consolidated in Figure 1, but it is necessary to go back and review each section individually, which hinders the understanding of the text. I think an appropriate answer is needed. Ideally, the diagram should change with each section. Excessive repetition of going back and forth from section to section is not desirable. Also, the experimental conditions (number of days of loading) are varied and the reason for these conditions is unclear.

Thank you for your constructive comments. As you rightly pointed out, we have restructured the figures accordingly. The experimental conditions have been placed at the beginning of each results figure (Figures 2a, 3a, 8a and 9a).

Regarding the number of days of IPP loading, we apologize for the lack of explanation. For the planar condition, we chose 3 days because the vascular endothelial cell network reaches its peak at 3 days and then begins to regress. On the other hand, for the cell sheet condition, the network peaks around 5 days, so we evaluated it 5 days. To clarify this point, we have added the following explanation to the “Results” section and included a representative image of both

environments on day 5 in “Supplementary Figure 2c”.

Results, page 9-10, lines 9-4:

“A promotion of the vascular endothelial network of the co-cultured vECs was more pronounced in the cell sheet environment than in the planar environment (Supplementary Fig. 2c). On day 3, The vascular endothelial network was compared between planar cultures and cell sheets (Supplementary Fig. 2d and 2e). As co-cultured cell sheets shrink immediately after detachment from the temperature-responsive culture dish, the analysis was conducted using the unit area “per mm²” to account for differences in cell density between the two groups. Both the number of network junctions and the length of the entire network were significantly greater in cell sheet environment than the planar culture conditions [cell sheet vs planar culture: (count/mm²); 12.4 ± 3.2 vs 128.1 ± 29.3 (Supplementary Fig. 2d), (mm/mm²); 3.9 ± 0.5 vs 17.2 ± 1.8 (Supplementary Fig. 2e)]. Furthermore, by day 5, the vascular network in planar cultures had deteriorated, whereas the vascular network in the cell sheet environment was well-maintained (Supplementary Fig. 2c). These results demonstrated that the cell sheet environment facilitated the formation of a dense vascular endothelial network and contributed to its sustained maintenance over 5 days.”

Results, page 10, lines 8-12:

“The aim of this experiment was to investigate the effect of IPP on the formation of a vascular endothelial network of co-cultured vECs in a cell sheet environment (Fig. 3a), because we have shown that the cell sheet is more likely to construct a dense vascular endothelial network. Furthermore, to observe the prolonged effects of IPP, we extended the evaluation period to 5 days, during which the vascular network can be maintained in the cell sheet environment.”

(2) The section "A cell sheet environment established vascular endothelial networks more effectively than a planar culture environment" in whole part mentioned to Supplementary Figures 1 and 2. These figures shouldn't be the main figure?

We appreciate your interest in the results regarding the vascular networks characteristic of the cell sheet environment. We also found them very intriguing and considered including them as a main figure. However, the main findings of this study focus on the effects of intermittent pneumatic pressure (IPP) on the co-culture of vECs and MSCs. Including these results as a main figure would deviate from the primary subject of the paper and could potentially confuse readers. Therefore, we concluded that presenting them as supplementary figures is more appropriate.

(3) It seems to me that the p-values obtained from the tests are conveniently used in different ways depending on the situation. Although the standard for whether there is a significant difference or not is given, there are many cases where the explanation is given as if there is a difference even in the case of a larger p-value. This is not appropriate for scientific discussion (especially in Figure 2).

Thank you for bringing this important point to our attention. Indeed, for the in vitro data in Figures 2d, 2e and 4h, it would be better to revise the interpretation of the statistical results and the description from the perspective of scientific discussion. On the other hand, as you know, in animal experiments, it is controversial to discuss results based solely on p-values. Regarding Figure 8j while the p-value is 0.13, the large effect size (Cohen's $d = 0.96$) suggests that we cannot definitively conclude there is no difference between the IPP (+) and IPP (-) groups. However, as you pointed out, it is clear that such an expression could lead to misunderstandings. Therefore, we have carefully revised the result description as follows:

These revisions did not affect the major findings of our paper.

Regarding **Fig. 2d and 2e**

Results, page 7, lines 3-4:

Before revision (original manuscript):

“Vascular endothelial cells in planar co-cultured conditions under intermittent positive pressure tend to better form vascular networks than in an unpressurized state.”

After revision (revised manuscript):

“Impact of intermittent positive pressure on endothelial cells forming vascular networks in planar co-culture conditions.”

Results, page 7, lines 7-13:

“Over a period of three days, GFP-HUVECs infiltrated the intracellular spaces and began to connect with each other, forming a vascular network **in both the IPP (-) and (+) groups (Fig. 2b, c). The number of network junctions did not differ significantly between the IPP (-) and IPP (+) groups [IPP (-) vs IPP (+) (count/mm²); 7.1 ± 4.1 vs 9.4 ± 2.7 , $p = 0.10$, Cohen's $d = 0.66$] (Fig 2d), and the total network length in the IPP (+) was also not significantly different [IPP (-) vs IPP (+) (mm/mm²); 3.5 ± 1.2 vs 4.2 ± 0.7 , $p = 0.09$, Cohen's $d = 0.90$] (Fig. 2e).”**

Discussion, page 20, lines 12-14:

Before revision (original manuscript):

“i) IPP, which was generated by an air compressing pressurization device, stimulated promotion of vEC network formation in a coculture condition of MSCs and vECs. Moreover, the network effect was even more pronounced in cell sheets”

After revision (revised manuscript):

“i) IPP, which was generated by an air-compressing pressurization device, **promoted vEC network formation in the coculture cell sheet of MSCs and vECs**”

Discussion, page 21, lines 10-13:

“The IPP method in the co-culture environment of the present study also **demonstrated a similar effect in promoting vascular network formation within a cell sheet structure**, despite clear differences in the pressure application methods.”

Regarding **Fig. 4h**

Results, page 12, lines 8-10:

Before revision (original manuscript):

“The proportion of GFP-HUVECs in all cells was slightly higher in the IPP (+) group ($6.2 \pm 1.0\%$) compared to the IPP (-) group ($4.3 \pm 1.5\%$) (Fig. 4h)”

After revision (revised manuscript):

“The proportion of GFP-HUVECs in all cells **was no significantly differences between** the IPP (+) group ($6.2 \pm 1.0\%$) and the IPP (-) group ($4.3 \pm 1.5\%$) (Fig. 4h)”

Regarding **Fig. 8**

Results, page 18, lines 6-7:

Before revision (original manuscript):

“**The in vivo transplanted cocultured cell sheet cultured under intermittent positive pressure produced better protein secretion into the rat systemic circulation.**”

After revision (revised manuscript):

“**Evaluation of the impact of intermittent positive pressure on the thickness and functionality of co-cultured cell sheets transplanted in vivo.**”

Results, page 18-19, lines 16-2:

“The intensity of GLuc luminescence in rat blood **tend to be** higher in the IPP (+) group [IPP (-) vs IPP (+) (AU); 2600 ± 224 vs 2824 ± 241 , $p = 0.13$, Cohen’s $d = 0.96$]: **Although the difference did not reach statistical significance ($p = 0.13$), the large effect size in the**

animal experience (Cohen's $d = 0.96$) suggests a notable trend (Fig. 8j)."

(4) The vascular network in Figure 2a, b and Figure 3a, b are low-magnification images, so it is difficult to understand whether the network is formed or not from the images. Please provide magnified images.

and

(5) Please also provide enlarged images of Figures 4c and 4e.

Thank you for your suggestions. To address your concerns, we have added magnified views to revised Figures 2b, 2c, 3b and 3c to better show the network formation and to Figure 4c and 4d to allow clearer observation of the details.

(6) Regarding the results of the oxygen concentration measurement (Figure 5), it was difficult to understand what is claimed to be a reproducible result. Also, why the 48-hour endpoint data? I think you should at least provide data for each day along with the other experimental conditions.

Thank you for your important suggestion regarding the results of the oxygen concentration measurements (Figure 5). As you pointed out, it may indeed be difficult to discern the differences in dissolved oxygen levels between the IPP and control groups based solely on the data at the 48-hour time point. However, our conclusions are based not only on this single time point but also on the overall trend of dissolved oxygen concentrations. To clarify our findings, we show you the graphs from three independent experiments, presented below. As demonstrated by these graphs, the dissolved oxygen concentration at the bottom of the culture medium was consistently higher in the IPP group, particularly after the first half-day of measurement.

Additionally, to further illustrate the effect of IPP on dissolved oxygen levels in the medium, we conducted a supplemental experiment measuring the oxygen content at the bottom of the medium in the absence of a cell sheet, as shown below and added as "Supplementary figure 3". This result consistently demonstrated higher oxygen levels in the IPP group. Based on these observations—both with and without cell sheets—we are confident that IPP increases dissolved oxygen levels in the culture medium.

Regarding your comment about the measurement period, we would like to explain the limitations we encountered. In our experience, data accuracy declines after three days of measurement due to sensor drift, requiring re-calibration. This limitation made it challenging

to collect reliable continuous data over five days. However, we understand your concern about relying solely on the 48-hour time point to present the differences in oxygen content. Therefore, we have analyzed data from three additional time points—12 hours, 24 hours, and 48 hours—and incorporated these results into Figure 5C for better clarity.

We will include these additional experimental data in the revised manuscript, as below:

Figure 5 C: analyzed data from three additional time points: 12 hours, 24 hours, and 48 hours

Supplementary figure 3: experiments measuring the oxygen content at the bottom of the medium in the absence of a cell sheet.

(7) In the section "The in vivo transplanted cocultured cell sheet cultured under intermittent positive pressure produced better protein secretion into the rat systemic circulation", it does not seem reasonable to draw such a conclusion based on the evaluation of GLuc luminescence emptiness alone.

As you suggested, our paper provides evidence supporting the statement “The in vivo transplanted cocultured cell sheet cultured under intermittent positive pressure produced better protein secretion into the rat systemic circulation” based solely on the luminescence intensity of GLuc, a secreted luminescent protein without physiological function. However, as you are aware, comparing the secretion capacity of transplanted cells is inherently challenging. This is because it is difficult to distinguish whether the detected proteins are secreted by the transplanted cells themselves or if the protein secretion levels of the recipient have increased as a result of the transplantation procedure. Therefore, in the present study, it is an important

finding in the field of cell transplantation that the transplanted cell sheet demonstrated differences in secretion between the IPP+ and IPP- groups, even when assessed using GLuc. This suggests that the serum concentrations of certain proteins secreted by the cell sheets may have increased.

However, to fully substantiate our claims, further studies, such as using genetic engineering to attach markers to the secreted proteins of transplanted cells, would be necessary. Accordingly, we have revised the descriptions in the Results section and added a statement as a limitation to indicate the need for future research.

We have revised the Introduction, Results and Discussion sections and added limitation to Discussion section, as follow:

Introduction, page 6, lines 3-4:

Before revision (original manuscript):

“Transplantation experiments also indicated increased protein secretion to the systemic circulation from IPP cultured grafts.”

After revision (revised manuscript):

“Transplantation experiments **suggest a potential improvement in protein secretion into the systemic circulation under IPP conditions.**”

Results, page 18, lines 6-7:

Before revision (original manuscript):

“**The in vivo transplanted cocultured cell sheet cultured under intermittent positive pressure produced better protein secretion into the rat systemic circulation.**”

After revision (revised manuscript):

“**Evaluation of the impact of intermittent positive pressure on the thickness and functionality of co-cultured cell sheets transplanted in vivo.**”

Discussion, page 20-21, lines 17-2:

Before revision (original manuscript):

“iii) The transplantation of a cell sheet cultured under IPP demonstrated successful engraftment while maintaining its thickness. Furthermore, a significant quantity of proteins was subsequently secreted after transplantation.”

After revision (revised manuscript):

“iii) The transplantation of a cell sheet cultured under IPP demonstrated successful engraftment while maintaining its thickness. **The results also suggested a potential**

improvement in certain protein secretion into the systemic circulation.”

Discussion, page 26, lines 8-12:

“Then, the results of the transplantation of cell sheets cultured under IPP suggest that the serum concentrations of certain secreted proteins also increase. However, to firmly demonstrate that the secretion of other functional proteins is also increased, further studies, such as using genetic engineering to attach markers to the secreted proteins of transplanted cells, would be needed.”

(8) Is the reason there are no error bars at all in the RNA-seq data because the variation is too small to see, or is there no variation at all?

Thank you for your insightful comment. In response, we have revised the RNA-seq graph by calculating Log₂ Fold Change (Log₂FC) values instead of Fold Change (FC) and added error bars based on the standard error of Log₂FC values calculated from each experiment. To ensure data transparency, we have also included a table in the manuscript that provides the raw CPM data for each sample. Furthermore, all data associated with this study have been uploaded to the Sequence Read Archive (SRA) under the project accession number PRJNA1163514. We greatly appreciate your feedback, which has allowed us to improve both the presentation and accessibility of our data.

We have revised the Figure 6 and supplementary Table 1, as described above, and along with the “Methods” and “Data availability statement” sections, as follow:

Method, page 36-37, lines 14-2:

“Log₂ fold changes were calculated from Generalized Linear Model (GLM), which corrects for differences in library size between the samples and effects of confounding factors. Log₂ fold changes were used to compare gene expression levels. Genes with all CPM values ≤ 0.5 were considered to have negligible expression and were labeled as “N.D.” (Not Detected) in the figures. For genes with CPM values of 0, a pseudo-value of CPM = 0.1 was assigned instead of 0 to enable the calculation of the standard error.”

Data availability statement, page 43, lines 8-11:

“All data supporting the findings of this study are included within the article (and its supplementary files). Additional data are available from the corresponding author upon reasonable request. All RNA-seq data have been uploaded to the Sequence Read Archive (SRA) under the project accession number PRJNA1163514.

(9) It is theoretically understandable that dissolved oxygen concentration changes during IPP incubation.

But what about the solubility of CO₂? Does the pH of the culture medium change? If the pH changes, this could affect the cellular response, please provide data on pH changes as well.

Thank you for bringing this important point to our attention. As you pointed out, pH changes are indeed an important factor influencing cellular response. Due to the limitations of our facility, we were unable to measure the dissolved CO₂ in the culture medium. Therefore, we focused on pH and conducted additional experiments. To account for the potential impact of cellular metabolism on pH, we conducted experiments to measure the pH of the culture medium under both IPP and atmospheric conditions (n=3). These measurements were performed without cells to eliminate the influence of cellular metabolism on pH. As below figure and raw data, the results showed that both groups had nearly the same pH values. There was no significant difference in the pH between day 1 and day 5 in either group. Based on these results, we conclude that even if there were differences in the dissolved CO₂ concentration under pressure, the pH changes would not be large enough to affect the cells.

The result's figures were added in Supplementary figure 3, along with O₂ dissolve experiments without cells.

	IPP(-)			IPP(+)		
	①	②	③	①	②	③
day0	7.5	7.5	7.5	7.5	7.5	7.5
day1	7.5	7.5	7.5	7.5	7.5	7.6
day3	7.5	7.5	7.6	7.5	7.5	7.6
day5	7.6	7.6	7.7	7.6	7.6	7.7

(10) The RNA-seq results show changes in HIF-1alpha, is this due to changes in dissolved oxygen concentration? If so, this would be a response to changes in oxygen concentration, not a response to pressure. Would similar results be obtained in a hyperoxic environment? Or what is the mechanism of the pressure response? I think it is necessary to organize the discussion on this point.

Thank you for your inquiry regarding the HIF-1alpha discussion. In this revision, we reanalyzed the RNA-seq data and modified the classification in Figure 6, creating a section for hypoxia-related genes. As shown in Fig 6d, other hypoxia-related genes (ARNT, ARNT2) besides HIF1alpha also decreased in IPP (+), suggesting that the decrease in HIF1-alpha is due to the increase in dissolved oxygen caused by IPP. Indeed, as you pointed out, experiments in a hyperoxic environment might support these results, but we have not conducted such experiments due to the limitation of not having a system for culturing in a hyperoxic environment. However, as discussed in “Major comment 6” and the decrease in the L/G ratio with IPP in Fig 2f, which indicating a preference for aerobic metabolism over IPP(-), are sufficient to demonstrate that the decrease in HIF1-alpha is due to dissolved oxygen. Furthermore, the additional experiments described below provide indirect evidence of the effect of IPP on increasing dissolved oxygen in the medium, which is consistent with the decrease in HIF1alpha being due to dissolved oxygen. We evaluated changes in cell sheet thickness under mild hypoxia (12% O₂) in both IPP (+) and IPP (-) groups. No difference in thickness was observed between IPP (+) and IPP (-) under mild hypoxia, and neither condition showed any thickness difference compared to IPP (-) under normoxia (Figure 4b and New Figure 7a). This indicates that the enhanced oxygen penetration under normoxia is likely a key factor in cell sheet thickening.

Results, page 16, lines 5-14:

“Intermittent positive pressure does not induce thickening of the co-cultured cell sheet under mild hypoxia

To investigate the effect of increased dissolved oxygen levels induced by IPP, we cultured co-cultured cell sheets under two conditions, IPP (+) and IPP (-), in a mild hypoxia environment (12% O₂, 5% CO₂), and compared their thickness. Under the mild hypoxia environment, no difference in cell sheet thickness was observed between the IPP (+) and IPP (-) conditions [IPP (-) vs IPP (+) (µm); 33.4 ± 3.5 vs 33.5 ± 3.3] (Fig. 7a). Furthermore,

there was no significant difference when compared to cell sheets cultured under normal conditions (Fig. 4b: IPP (-)), indicating that the thickening effect of IPP on cell sheets is abolished under mild hypoxia conditions.”

(11) I think it is weak to discuss the promotion of vascular endothelial network formation only by the fluctuation of RNA expression. If possible, the authors should verify the expression of representative proteins involved in the promotion of network formation among those that fluctuated.

Thank you for your recommendation. As you suggested, the discussion on the promotion of vascular endothelial network formation solely based on the fluctuation of RNA expression is indeed a weakness. However, the primary focus of our study is to highlight the role of shear stress induced by IPP in promoting angiogenesis. To address this, we performed additional experiments focusing on shear stress and its effects on vascular network formation. Specifically, we conducted eNOS inhibition experiments using the inhibitor L-NAME, which targets NOS3, a gene associated with shear stress signaling activated by IPP (Figure 6a). Our findings revealed that inhibiting eNOS did not affect the vascular network under IPP (-) conditions. However, under IPP (+) conditions, vascular network formation was significantly suppressed to levels comparable to those observed under IPP (-) (New Figure 7b, c). These results suggest that shear stress induced by IPP plays a critical role in the construction of vascular networks.

In addition, regarding the vascular paracrine factors associated with angiogenesis that you mentioned, we plan to conduct further experiments. These will include a comprehensive protein analysis of the culture supernatant to better elucidate their roles in vascular network formation. We have revised the Results and Discussion sections, as follow:

Added new figure as Figure 7b, c

Results, page 17-18, lines 3–4:

“eNOS inhibition suppresses vascular network formation in co-cultured cell sheets under intermittent positive pressure

To investigate the effect of IPP on vascular endothelial network formation, we added the eNOS inhibitor L-NAME to the co-cultured cell sheets and compared vascular endothelial network formation under IPP (+) and IPP (-) conditions (Fig. 7b, c). Treatment with L-NAME under IPP (-) conditions did not lead to any changes in the formation of the vascular endothelial network compared to the group cultured under IPP(-) conditions without L-NAME [IPP (-) L-NAME (-) vs IPP (-) L-NAME (+); Total number of endothelial network junctions (count/mm²): 60.2 ± 16.6 vs 59.5 ± 14.6, Total length of the endothelial network (mm/mm²): 5.7 ± 0.8 vs 5.7 ± 0.7]. In contrast, when the cell sheets were cultured under IPP (+) conditions, the addition of L-NAME resulted in a suppression of vascular network formation compared to the group without L-NAME [IPP (+) L-NAME (-) vs IPP (+) L-NAME (+); Total number of endothelial network junctions (count/mm²): 84.1 ± 11.7 vs 67.4 ± 17.0, Total length of the endothelial network (mm/mm²): 6.7 ± 0.5 vs 6.1 ± 0.6]. Furthermore, no significant differences in vascular network formation were observed when comparing the L-NAME-treated cell sheets cultured under IPP(-) and IPP(+) conditions [IPP (-) L-NAME (+) vs IPP (+) L-NAME (+); Total number of endothelial network junctions (count/mm²): 59.5 ± 14.6 vs 67.4 ± 17.0, Total length of the endothelial network (mm/mm²): 5.7 ± 0.7 vs 6.1 ± 0.6] (Fig. 7b, c). Inhibition of eNOS by L-NAME suppressed the vascular network formation induced by IPP (+), indicating the involvement of eNOS in the process.”

Discussion, page 22, lines 5-15:

“In the present experiment, the expression of VE-cadherin increased, as did the expression of multiple related genes downstream of VE-cadherin signals (VEGFR2 gene KDR, eNOS gene NOS3, and cyclin A gene CCNA1) caused by shear stress¹⁵. Furthermore, eNOS inhibition experiments in the present study demonstrated that inhibiting eNOS under IPP (+) conditions significantly suppressed vascular network formation, bringing it to levels comparable to those observed under IPP (-) conditions. This result is consistent with the RNA-seq findings and suggests a strong correlation between vascular network formation induced by the IPP method and shear stress. In contrast, the increased expression of bFGF³⁰, TRPV4³¹, integrin α 5³², AQP1, α 1-AR, and SR-2A¹³, which have all been reported to be associated with hydrostatic stimulation, was not confirmed in the present study.”

(12) I strongly recommend reconsideration of the assay method. I do not think it is appropriate to use the Student's t-test in a comparative test between groups of samples where equal variances cannot be assumed. Looking at the data, I do not think that equal variances can be guaranteed.

Thank you for highlighting your concerns regarding the statistical methods employed in our

analysis. We appreciate the opportunity to clarify and address this point.

In our original manuscript, we stated that Student's t-tests were used throughout; however, this was not entirely accurate. In fact, paired t-tests were applied where appropriate. To ensure the robustness of our analysis, we have now re-evaluated all datasets for normality and equal variances.

Normality was assessed using the Shapiro-Wilk test with a significance threshold of 0.05, and equal variances were evaluated using the F-test, also with a threshold of 0.05. Furthermore, we carefully reviewed the nature of each comparison. For Figures 8j and 8l, the comparisons were made between different animals and were therefore treated as unpaired data. In contrast, all other comparisons involved samples derived from the same cell population under different conditions, and these were analyzed as paired data.

For the paired data, normality was confirmed for most figures, and paired t-tests were used. However, for Figure 2h, 4e, 4h, 7b and 7c, since normality was not observed, we applied the Wilcoxon signed-rank test.

For the unpaired data in Figure 8j, both normality and equal variances were confirmed, and we therefore used the Student's t-test. Conversely, for Figure 8l, normality was not observed, and we applied the Mann-Whitney U test as a result.

We have significantly revised the manuscript to include these details and the reference to effect size as pointed out in your minor comment (3), in order to ensure accurate reporting of the statistical methods used, as outlined below. We hope this clarification addresses your concerns and thank you for bringing this important point to our attention.

Method, page 41-42, lines 13-11:

“Statistical analysis

All statistical analyses were conducted using appropriate tests based on the characteristics of the data. For comparisons involving the same cell population under different culture conditions, data were treated as paired. For comparisons between different animals (Figures 8j and 8l), data were treated as unpaired. Normality of the data was assessed using the Shapiro-Wilk test, with a significance threshold of 0.05. Equal variances were evaluated using the F-test, also with a significance threshold of 0.05. For paired datasets, normality was confirmed for all figures except for Figure 2h, 4e, 4h, 7b and 7c. Paired t-tests were applied to datasets with normal distribution. For Figures 2h, 4e, 4h, 7b and 7c which did not meet the normality, the Wilcoxon signed-rank test was used. For unpaired datasets, Figure 8j met both normality and equal variances, allowing the use of the Student's t-test. In contrast, Figure 8l did not meet the normality assumption, and therefore, the Mann-Whitney U test was applied. The effect size was calculated using Cohen's d to evaluate the magnitude of differences between IPP (-) and

IPP (+). The interpretation of Cohen's d followed the conventional thresholds: 0.2 indicates a small effect size, 0.5 indicates a moderate effect size, and values greater than 0.8 indicate a large effect size⁴⁴. These statistical methods were chosen to ensure the appropriate analysis of the data based on their distribution and variance properties.”

References, No. 44:

“44. Cohen, J. *Statistical Power Analysis for the Behavioral Sciences*. (Routledge, 2013). doi:10.4324/9780203771587.”

Minor points

(1) *The PDF file of the manuscript is garbled, with missing text in several places (around lines 1~20 and around lines 480~580). The author's name is missing.*

(2) *The number of pages is not displayed correctly.*

Thank you for bringing these inadvertent omissions to our attention. We believe these issues arose during the automatic PDF conversion process at the time of submission. We have carefully reviewed the file generated after this conversion and submitted it accordingly.

(3) *In the test, the effect size should be reported along with the P-value calculation.*

Thank you for the helpful suggestion. I have performed Cohen's d test and included it along with the P-value in the figure legend (Figure 2,3,4,7, 8 and 9, Supplementary Table 1). This additional analysis did not require any changes in the interpretation of the results.